# Low-Density Lipoprotein Cholesterol-Lowering Drugs: A Narrative Review

**DOI:** 10.3390/jcm13040943

**Published:** 2024-02-07

**Authors:** Nicola Ferri, Massimiliano Ruscica, Sergio Fazio, Alberto Corsini

**Affiliations:** 1Department of Medicine (DIMED), University of Padova, 35122 Padova, Italy; 2Veneto Institute of Molecular Medicine (VIMM), 35129 Padua, Italy; 3Department of Pharmacological and Biomolecular Sciences “Rodolfo Paoletti”, University of Milan, 20133 Milan, Italy; alberto.corsini@unimi.it; 4Department of Cardio-Thoracic-Vascular Diseases, Foundation IRCCS Cà Granda Ospedale Maggiore Policlinico, 20122 Milan, Italy; 5Regeneron Pharmaceuticals Inc., Tarrytown, NY 10591, USA; sergio.fazio@regeneron.com

**Keywords:** LDL-cholesterol, PCSK9, statins, bempedoic acid, ezetimibe, inclisiran

## Abstract

The modern history of cholesterol-lowering drugs started in 1972 when Dr. Akira Endo identified an active compound (compactin) that inhibited cholesterol biosynthesis from the culture broth of blue–green mold (*Penicillium citrinum* Pen-51). Since 1987, statins have represented the milestone for the treatment of atherosclerotic cardiovascular disease. A new therapy for the treatment of hypercholesterolemia since the discovery of statins is ezetimibe, the first and only agent inhibiting intestinal cholesterol absorption. Ezetimibe was approved by the FDA in October 2002. A year later, the association between gain-of-function PCSK9 genetic mutations and hypercholesterolemia was reported, and this discovery opened a new era in lipid-lowering therapies. Monoclonal antibodies and small-interfering RNA approaches to reduce PCSK9 were developed and approved for clinical use in 2015 and 2022, respectively. Finally, the newly approved bempedoic acid, an oral adenosine triphosphate citrate lyase inhibitor that lowers LDL-C, is able to reduce major adverse cardiovascular events in both primary and secondary prevention. In the present narrative review, we summarize the pharmacological properties and the clinical efficacy of all these agents currently used for a tailored therapy of hypercholesterolemia in patients with atherosclerotic cardiovascular disease.

## 1. Introduction

Atherosclerotic cardiovascular disease (ASCVD) is the leading cause of death worldwide, and dyslipidemia is one of the primary causal risk factors [1]. In contemporary cardiovascular care, almost all patients with or at a high risk of developing atherosclerosis are treated with statin therapy, the first-line recommended treatment for lowering low-density lipoprotein cholesterol (LDL-C) [1]. This evidence is sustained by the results of a meta-analysis comprising data from 26 randomized clinical trials (RCTs) showing a linear association between a reduction in LDL-C and ASCVD morbidity and mortality, in both primary and secondary prevention settings [2]. However, since the most recent guidelines recommend reducing LDL-C to levels below 55 mg/dL (1.4 mmol/L) in very high-risk patients, and even lower (<40 mg/dL; <1.0 mmol/L) for patients with ASCVD who experience a second vascular event within 2 years, to mitigate the ASCVD risk [3], statin monotherapy is not sufficient for many patients. Considering that the achieved cardiovascular benefit is independent of the mechanism by which LDL-C is lowered [4], it is important to shift the treatment paradigm for very high-risk patients from “an intensive statin therapy” approach to an “intensive lipid-lowering therapy” approach [5].

A major therapeutic boost in this field came from the broader availability of non-statin alternatives, such as ezetimibe and proprotein convertase subtilisin/kexin type 9 (PCSK-9) inhibitors (both monoclonal antibodies and small interfering RNA) [6]. Very recently, the cardiovascular benefit of bempedoic acid, which lowers LDL-C upstream from statins, has been proven in patients with statin intolerance [7] (Figure 1). This narrative review discusses established and newly available LDL-lowering therapies for use in ASCVD prevention.

## 2. Statins

Statins, or 3-hydroxy-3-methylglutaryl-CoA (HMG-CoA) reductase inhibitors, are small molecules that inhibit the synthesis of endogenous cholesterol [8]. They represent the result of research that began in the 1970s through the screening of more than 6000 microbial strains that led to the discovery of mevastatin, a competitive inhibitor of HMG-CoA reductase [9,10]. The first generation of statins (lovastatin, simvastatin, and pravastatin) was, therefore, obtained by fermentation, while fluvastatin, atorvastatin, rosuvastatin, and cerivastatin are the second generation of statins, obtained by synthesis. Currently, there are seven statins available for clinical use: lovastatin, simvastatin, fluvastatin, atorvastatin, pravastatin, rosuvastatin, and pitavastatin.

### 2.1. Mechanism of Action

The mechanism of action, common to all statins, is based on competitive inhibition of the enzyme HMG-CoA reductase, a rate-limiting step of cholesterol synthesis in all cells, but mostly targeting the hepatocytes, due to their high hepatic extraction [8,11]. Statins, by inhibiting this enzyme, reduce the de novo cholesterol synthesis, thus activating the sterol-dependent processing of the sterol-regulatory binding protein-2 (SREBP-2) pathway and increasing the level of the LDL receptor, which results in a higher hepatic uptake of apolipoprotein (apo)B-containing lipoproteins and a plasma lipid-lowering effect [12]. 

### 2.2. Pharmacodynamics

Statins reduce LDL-C levels by 20–60% in a dose-dependent manner, with different potency and efficacy (Figure 2) [13,14]. These drugs slightly reduce triglyceride levels (10–30%), but only in subjects with hypertriglyceridemia (i.e., the greater the LDL-C reducing efficacy, the more effective the triglycerides reduction) [15]. In addition to these actions, statins reduce apoB, non-HDL cholesterol, very low-density lipoprotein cholesterol (VLDL-C), and VLDL triglycerides, and increase apoA-I and high-density lipoprotein cholesterol (HDL) by 5–10% [3]. Rosuvastatin is the most potent drug of this class, producing a reduction of LDL-C of more than 50% (55–60% at 40 mg), followed by atorvastatin (45–50% at 80 mg) and pitavastatin (45–50% at 4 mg) [16]. Finally, it should be mentioned that lipoprotein(a) levels increase in patients who were started on statin therapy [17]. Specifically, the mean percentage change from baseline ranged from 11.6% to 20.4% in the case of pravastatin and from 18.7% to 24.2% in the case of atorvastatin [18].

Reproduced with permission from Catapano, A.L. et al. [16]. Atorva: atorvastatin; fluva: fluvastatin; prava: pravastatin; simva: simvastatin; rosuva: rosuvastatin; pitava: pitavastatin.

### 2.3. Pharmacokinetics

The pharmacokinetic characteristics of statins are summarized in Table 1 [11]. After oral administration, statins undergo significant hepatic extraction that negatively impacts their bioavailability due to a catabolism involving CYP3A4 for lovastatin, simvastatin, and atorvastatin and by CYP2C9 for fluvastatin and rosuvastatin [19]. CYP3A4 metabolism may determine significant drug interaction with potent cytochrome inhibitors or inducers, while fluvastatin and rosuvastatin plasma concentrations could be altered in patients with single nucleotide polymorphisms (SNP) of CYP2C9 [20]. A significant proportion of rosuvastatin and pravastatin dose is excreted with urine by the kidneys [19]. Importantly, atorvastatin, rosuvastatin, and pitavastatin have a long half-life elimination time and can be administered in the morning, while simvastatin, fluvastatin, and pravastatin, due to their short half-life, should be taken in the evening, when the hepatic expression of HMG-CoA reductase is highest.

Considering the pharmacokinetic properties of statins, kidney disease does not affect the plasma concentrations or lipid-lowering effects of hepatically eliminated statins (e.g., atorvastatin), while caution is required for statins with higher renal clearance (rosuvastatin and pravastatin). Plasma concentrations of atorvastatin, and its active metabolites, appear markedly increased (approximately 16-fold C_max_ and approximately 11-fold AUC) in patients with chronic liver disease (Child–Pugh B). In a study of subjects with varying degrees of hepatic impairment, there was no evidence of increased systemic exposure to rosuvastatin in subjects with a Child–Pugh score ≤ 7; two subjects with Child–Pugh scores of 8 and 9 had at least a 2-fold increase in systemic exposure to rosuvastatin [21].

### 2.4. Safety

Statins are safe and well-tolerated, although some patients may develop muscular complications, type 2 diabetes, and increased levels of liver aminotransferases [11]. An analysis of >4 million patients reported that the overall prevalence of statin intolerance was 9.1% (95%CI 8.0–10%) [22], with no differences based on lipid solubility (4.0% (2.0–5.0%) vs. 5.0% (4.0–6.0%)) [23]. In particular, statin-associated muscle symptoms (SAMS) represent the most common symptoms that result in statin intolerance or statin discontinuation, increasing the risk for recurrent myocardial infarction (50%) and coronary heart disease events (51%) [24]. Specifically, statin intolerance is defined as one or more adverse effects associated with statin therapy, which resolves or improves with dose reduction or discontinuation. It can be classified as a complete inability to tolerate any dose of a statin, or partial intolerance, with an inability to tolerate the dose necessary to achieve the patient-specific therapeutic objective. To classify a patient as having statin intolerance, a minimum of two statins should have been attempted, including at least one at the lowest approved daily dosage [25]. However, data extrapolated from randomized clinical trials and real-world studies are contrasting on this issue (i.e., in long-term clinical trials, muscle adverse events occur at similar rates, regardless of allocation to a statin or placebo). Indeed, trials with statins excluded many patients who would be prescribed a statin in clinical practice. These trials excluded patients unlikely to adhere to the demands of a long-term study and those taking known interacting drugs [26]. On this matter, an analysis of 19 placebo-controlled trials concluded that >90% of all reports of muscle symptoms by participants allocated statin therapy were not due to the statin [27]. Conversely, in routine care settings, this percentage can increase until 30% [28,29]. However, rechallenge is possible, by changing statin type and reducing the dose. As established by both interventional studies and clinical practice, it is recommended that before rechallenging, a washout period may be useful to allow symptom resolution, which varies among individuals [30]. A further approach can be alternate-day dosing by prescribing atorvastatin and rosuvastatin which possess a long half-life [31].

Concerning new-onset diabetes, although statin therapy is associated with a 9% increased risk, it is low compared with the reduction in coronary events. Indeed, the treatment of 255 patients with statins for 4 years resulted in one extra case of diabetes [32,33]. Regarding liver toxicity, the incidence of transaminase elevations >3-fold the upper limits of normal (ULN) have never been reported. This effect seems dose-related with a temporary pattern and, thus, clinicians should not be discouraged to prescribe statins even when a mild rise of serum liver enzyme is present [34].

The development of cataracts or cognitive dysfunction and several more adverse events reported in the literature (e.g., proteinuria and hematuria) have never been confirmed to have a causal link with statin use [11]. The prevalence of complete statin intolerance might often be overestimated, highlighting the need for the careful assessment of patients with potential symptoms related to statin intolerance. The excess rate of SAMS mainly derives from the so-called nocebo effect, as demonstrated in the ASCOT-LLA (Anglo-Scandinavian Cardiac Outcomes Trial—Lipid Lowering Arm) study. During the non-blinded, non-randomized phase of this trial, SAMS were reported at a significantly higher rate by participants taking statins than by those who were not (Hazard ratio (HR) = 1.41 (95%CI 1.10–1.79)) [35]. Indeed, muscle symptoms are referred more often when patients and their doctors are aware that statin therapy has been used and not when its use was blinded. 

### 2.5. Drug Interaction

Statins, and all lipid-lowering therapies, are chronically used drugs and thus likely to undergo drug interaction in poly-treated subjects. Approximately half of all drugs currently available in clinical practice are metabolized by CYP450 3A4, and so are some statins (simvastatin, lovastatin, cerivastatin, and atorvastatin) [20]. Thus, statins are susceptible to drug interactions when co-administered with potential inhibitors of this enzyme (for specific reviews on this topic, please refer to [36,37]). Indeed, myositis and rhabdomyolysis have been reported following the concurrent use of simvastatin or lovastatin and cyclosporine A, azole antifungals (ketoconazole, itraconazole), protease inhibitors (ritonavir), and some macrolides antibiotics, including erythromycin, azithromycin, and clarithromycin [37]. 

Considering hypolipidemic drugs, gemfibrozil, when combined with any statins and especially with cerivastatin, determined an increased risk of rhabdomyolysis [38]. Gemfibrozil is also known to reduce the glucuronidation and elimination of statins [36]. On the contrary, cases of myopathy are limited when statins are associated with fenofibrate, bezafibrate, or ciprofibrate [3]. More intriguingly, statins, ezetimibe, and fenofibrate have been shown to reduce the exposure of alirocumab by approximately 40%, 15%, and 35%, respectively. A similar effect was observed with simvastatin and evolocumab. This clinically irrelevant drug interaction is due to the induction of PCSK9 expression by statins and fibrates that accelerates the target-mediated clearance of alirocumab and evolocumab [39].

The IL-6 receptor inhibitor tocilizumab may reverse the inhibitory effect of this cytokine on CYP450 expression. Indeed, IL-6 down-regulates the expression of CYP3A4, 2B6, and 2C8 mRNA. A pharmacokinetic study conducted in patients with rheumatoid arthritis demonstrated that C_max_ and AUC of simvastatin and its main metabolite (β-hydroxy-simvastatin acid) were significantly reduced after a single infusion of tocilizumab [40].

In patients with non-valvular atrial fibrillation in therapy with dabigatran, a direct oral anticoagulant, the co-administration of simvastatin and lovastatin has been associated with a higher risk of major hemorrhage relative to other statins [41]. This interaction could be related to the inhibitory action of simvastatin and lovastatin on P-gp that may lead to an increased exposure of dabigatran and a higher risk of hemorrhagic stroke. No clinically relevant DDIs have been reported with statins and apixaban, edoxaban, and rivaroxaban [42]. Similarly, the dose of simvastatin should not exceed 20 mg per day when administered with the antiarrhythmic P-gp inhibitor amiodarone [36].

The antiplatelet drug ticagrelor is a weak inhibitor of CYP3A4 and P-gp and has been shown to increase atorvastatin C_max_ and AUC by 23% and 36%, respectively, and simvastatin C_max_ and AUC by 81% and 56%, respectively [43,44]. When prescribed with ticagrelor, the dose of simvastatin and lovastatin should not exceed 40 mg daily. No dosing restrictions are needed in the case of combination with the other statins and no clinically significant interaction has been reported with clopidogrel and prasugrel [45].

Finally, the combination of a neprilysin inhibitor, sacubitril, with the angiotensin receptor blocker valsartan has been shown to double atorvastatin C_max_ and AUC, although no clinically significant adverse effects were reported [46]. This interaction can be due to the inhibition by sacubitril of organic anion transporting polypeptides (OATP)1B1, OATP1B3, OAT1, and OAT3 transporters that recognize atorvastatin, as well as other statins. Currently, no dose adjustments are required for statins co-administered with sacubitril/valsartan [45], although additional pharmacokinetics studies are needed to further unravel this issue.

### 2.6. Clinical Efficacy

Statins are among the most widely used drugs worldwide and reduce LDL-C on average by 30–50% [16]. Statin-driven ASCVD benefit has been proven by a number of large interventional studies and meta-analyses [3]. They can reduce the risk of ASCVD events by approximately 30%, with an additional 15% reduction with high-intensity compared to low- or moderate-intensity regimens [47,48]. Large-scale evidence from randomized trials shows that statin therapy reduces the risk of major vascular events (i.e., coronary deaths or myocardial infarctions, strokes, and coronary revascularization procedures) by approximately one-quarter for each 38.7 mg/mL reduction in LDL-C during each year (after the first) that it continues to be taken [49]. Thus, a larger absolute benefit would accrue with more prolonged therapy, and these benefits persist long term. A meta-analysis of data from 27 trials of statin therapy showed all-cause mortality reductions with statin therapy regardless of sex, namely, relative risk (RR) 0.91 (99%CI 0.84–0.99) in women and RR 0.90 (99%CI 0.86–0.95) in men [50]. However, real-life is a field fraught with uncertainties. Despite guideline recommendations, among patients with established ASCVD, in the US, fewer than one-half are statin users and only 22.5% are on a high-intensity formulation [51]. In addition, in the case of end-stage renal disease, statins have shown no benefit on cardiovascular outcomes. Briefly, the results of the AURORA (A study to evaluate the Use of Rosuvastatin in subjects On Regular hemodialysis: an Assessment of survival and cardiovascular events) study showed that rosuvastatin had no effect on nonfatal myocardial infarction, nonfatal stroke, or death from cardiovascular causes (HR = 0.96; 95%CI 0.84 to 1.11) [52]. The 4D (Die Deutsche Diabetes Dialyse) study conducted on patients undergoing hemodialysis found the non-superiority of atorvastatin in reducing the composite primary end point of cardiovascular death, nonfatal myocardial infarction, and stroke [53].

Finally, it becomes of interest to evaluate the impact of statins in patients living with HIV. These people have an excess 1.5–2.2-fold risk of myocardial infarction (MI) compared to HIV-uninfected individuals with a similar CV risk profile. Specifically, HIV people carry an increased non-calcified and high-risk vulnerable plaque. In this context, the REPRIEVE (Evaluating the Use of Pitavastatin to Reduce the Risk of Cardiovascular Disease in HIV-Infected Adults) study has demonstrated that people with HIV infection who received pitavastatin had a lower risk of major adverse cardiac events (MACE) than those who received the placebo over a median follow-up of 5.1 years [54]. 

## 3. Ezetimibe

Ezetimibe was first designed as a potential ACAT (acylcoenzyme A cholesterol acyltransferase) inhibitor and was subsequently discovered to reduce cholesterol intestinal absorption. Thus, ezetimibe was identified without a clear understanding of its molecular target, which was also unknown at the time [55]. Thus, ezetimibe is a selective inhibitor of intestinal cholesterol absorption [55].

### 3.1. Mechanism of Action

Ezetimibe is the first in a class of cholesterol-lowering agents that acts by inhibiting Niemann–Pick C1-Like 1 (NPC1L1) and thus intestinal cholesterol absorption [56]. Ezetimibe targets the small intestine and inhibits the absorption of cholesterol, resulting in a decrease in the passage of intestinal cholesterol to the liver. In humans, 2 weeks of treatment with 10 mg of ezetimibe determined a reduction of 54% of fractional cholesterol absorption rates [57]. In addition, cholesterol synthesis increased by 89%, suggesting that intestinal cholesterol absorption and hepatic cholesterol production are linked by a common sensing loop [57]. 

### 3.2. Pharmacodynamics

In monotherapy, 2 weeks of treatment with ezetimibe determined a mean percentage change from baseline for LDL and total cholesterol of −20.4% and −15.1%, respectively, whereas absorption markers such as campesterol and sitosterol were decreased by 48% and 41%, respectively [57]. The two distinct mechanisms of action of statins (inhibition of cholesterol biosynthesis) and ezetimibe (reduction of cholesterol absorption) together produce a complementary reduction in cholesterol. Co-administration of ezetimibe with statins leads to a further reduction in LDL-C, by 18–20%, regardless of the dose and type of statin used [58]. Overall, this therapeutic approach leads to a LDL-C reduction similar to the one obtained with high-intensity statins (i.e., 40 mg rosuvastatin or 80 mg atorvastatin) (Figure 3) [13]. The drug has been shown, in combination with simvastatin, to reduce the risk of cardiovascular events in relation to the reduction of LDL-C, both in nephropathic and in acute coronary syndrome patients [59,60].

Atorva: atorvastatin; Eze: ezetimibe; Simva: simvastatin; Rosuva: rosuvastatin. Reproduced with permission from Ferri, N. et al. [61].

### 3.3. Pharmacokinetics

After oral administration, ezetimibe is rapidly absorbed and extensively conjugated to the pharmacologically active glucuronide derivative. Mean C_max_ is observed within 1–2 h for ezetimibe–glucuronide and 4–12 h for ezetimibe [62]. The oral bioavailability of ezetimibe was similar under both fed and fasting conditions, and thus the drug can be administered with or without meals [62]. Ezetimibe and ezetimibe–glucuronide are 99.7% and 88–92% bound to human plasma proteins, respectively [63].

Both ezetimibe and ezetimibe–glucuronide are slowly eliminated from plasma, with significant enterohepatic recirculation [64]. Following oral administration of ^14^C-ezetimibe (20 mg) to humans, approximately 93% of the total radioactivity reached the circulation, and approximately 78% and 11% of the administered radioactivity was recovered in the feces and urine, respectively [65]. The mean AUC for total ezetimibe increased approximately 1.7-fold in patients with mild hepatic impairment (Child–Pugh score 5 or 6) compared to healthy subjects. In a 14-day multiple dose (10 mg/day) study in patients with moderate hepatic impairment (Child–Pugh score 7 to 9), the mean AUC for total ezetimibe increased approximately 4-fold on day 1 and at day 14 compared to healthy subjects. No dose adjustment is required in patients with mild hepatic impairment [66]. Due to the unknown effects of increased exposure to ezetimibe in patients with moderate or severe hepatic impairment (Child–Pugh score > 9), ezetimibe is not recommended for these patients [66]. Notwithstanding, ezetimibe produces significant and progressive reductions in plasma plant sterol concentrations in patients with sitosterolemia, consistent with the hypothesis that ezetimibe inhibits the intestinal absorption of plant sterols as well as cholesterol, leading to reductions in plasma concentrations [67].

After a single 10 mg dose of ezetimibe in patients with severe renal disease (CrCl ≤ 30 mL/min/1.73 m^2^), the mean AUC for total ezetimibe increased by approximately 1.5 times compared to healthy subjects [66]. This result is not considered clinically significant. No dosage adjustment is necessary for patients with renal impairment.

### 3.4. Safety

The tolerability of ezetimibe therapy has been evaluated in several clinical studies. In controlled studies vs. the placebo, ezetimibe was well tolerated, with no higher incidence of adverse reactions than the placebo [68]. Subjects treated with a statin and ezetimibe have an incidence of transaminase elevations > 3 times the ULN than subjects receiving a statin as monotherapy [68]. An increase in creatine kinase (CK) values > 10-fold above the ULN was a rare event and completely comparable in incidence between the ezetimibe and combined therapy groups [68].

Abdominal pain and diarrhea were reported in approximately 3% of patients; similarly, arthralgia was reported in 3.8% of patients. Both side effects were not significantly different from the placebo.

### 3.5. Drug Interaction

No clinically significant pharmacokinetic interactions have been observed between ezetimibe and drugs known to be metabolized via CYP1A2, 2C8/9, 2D6, and 3A4. Ezetimibe does not induce or inhibit CYP450. The concomitant use of ezetimibe and the immunosuppressant cyclosporine A in transplant subjects should be closely monitored, as the concentration of ezetimibe was shown to be increased 12-fold [63], while ezetimibe does not alter cyclosporine exposure [69]. The disposition of ezetimibe in humans is determined by expression of intestinal UGT1A1, P-gp, and MRP2 [70]. Co-administration with rifampin leads to reduced effects of ezetimibe caused by faster elimination of ezetimibe via glucuronidation and subsequent intestinal or hepatic secretion via P-gp and MRP2 [70].

### 3.6. Clinical Efficacy

The Improved Reduction of Outcomes: Vytorin Efficacy International Trial (IMPROVE-IT) randomized 18,144 patients hospitalized for ACS in the previous 10 days to either ezetimibe or the placebo, in addition to simvastatin (40 mg/die) [59]. The difference between treatments in LDL-C was 16.2 mg/dL (22.4% relative difference). The composite of cardiovascular death, MI, unstable angina requiring rehospitalization, coronary revascularization ≥ 30 days after randomization, or stroke (namely, the primary endpoint) was reduced with ezetimibe by 2.0% (HR = 0.936; 95%CI 0.89–0.99) [59]. Adding ezetimibe to the statin consistently reduced the risk for cardiovascular events in post-ACS patients irrespective of baseline LDL-C values [71]. However, in patients aged ≥ 75 years, ezetimibe, as a monotherapy, was able to reduce, compared to usual care, a composite of sudden cardiac death, fatal/nonfatal myocardial infarction, coronary revascularization, or fatal/nonfatal stroke by 34% [72].

## 4. Monoclonal Antibodies Blocking PCSK9 Action

In 2003, proprotein convertase subtilisin kexin type 9 (PCSK9) was discovered as the third gene involved in autosomal dominant hypercholesterolemia [73]. PCSK9 is secreted by the liver and acts as a negative regulator of the number of cell surface LDL receptors in hepatocytes. This molecular target can be inhibited by monoclonal antibodies (mAbs) such as alirocumab and evolocumab. Alirocumab is a fully human IgG1 protein produced in Chinese Hamster Ovary (CH) cells by recombinant DNA technology, while evolocumab is a fully human IgG2 mAb.

### 4.1. Mechanism of Action

Alirocumab and evolocumab bind selectively to circulating PCSK9, preventing its interaction with the LDL receptor expressed on the hepatocytes, thus avoiding PCSK9-mediated LDL receptor degradation [74]. Increased hepatic LDLR levels result in a higher liver uptake of serum LDL-C. Importantly, the onset of action of mAbs occurs within 4 h after subcutaneous administration, with 97% of plasma PCSK9 bound to antibody and unable to interact with the LDL receptor [75]. Of note, total plasma levels of PCSK9 increase in response to its inhibition by blocking antibodies, as the LDL receptor also represents the main elimination route of PCSK9 [76].

### 4.2. Pharmacodynamic

By inhibiting PCSK9, both mAbs produce a sustained and effective reduction of LDL-C (≈55%) and total cholesterol (≈35%) [77,78]. Thanks to their efficacy, both in monotherapy and in combination with statins and other lipid-lowering drugs (ezetimibe and bempedoic acid) [8], they have become the most effective therapies for the treatment of severe and familial hypercholesterolemia (FH). Evolocumab and alirocumab significantly reduced apoB (≈50%), non-HDL cholesterol (≈50%), lipoprotein(a) (≈25%), and triglycerides (≈15%) [77,78], but they did not modify levels of high-sensitivity C reactive protein (hsCRP) [79]. 

### 4.3. Pharmacokinetics

Following a single subcutaneous (s.c.) administration of mAb anti-PCSK9 in healthy volunteers, maximum serum concentrations were achieved in 3 to 4 days [80]. The bioavailability of alirocumab and evolocumab is approximately 70%. The mean steady-state volume of distribution is 3 L, suggesting limited tissue distribution. Both antibodies are glycosylated proteins similar to native immunoglobulins and are not eliminated by hepatic metabolism mechanisms. The metabolism and elimination of the drug is expected to follow the immunoglobulin clearance pathways, resulting in degradation to small peptides and individual amino acids by the reticuloendothelial system [39]. Evolocumab has an estimated effective half-life of 11–17 days and alirocumab has an estimated effective half-life of 12 days (Table 2) [80].

No dose adjustment is required in patients with renal impairment. After s.c. administration, the pharmacokinetic profiles of alirocumab and evolocumab in subjects with mild and moderate hepatic impairment were similar compared to subjects with normal hepatic function. There are no data available for patients with severe hepatic impairment [81].

The recommended dose of evolocumab is 140 mg every 2 weeks or 420 mg monthly [81]; the starting dose of alirocumab is 75 mg, administered subcutaneously once every 2 weeks, which can be increased to 150 mg once every 2 weeks or 300 mg once every 4 weeks [82].

### 4.4. Safety

The most common adverse reactions reported in phase 3 clinical trials were muscle pain, nasopharyngitis, back pain, upper respiratory tract infections, headache, joint pain (7%), flu-like symptoms, or an increase in alanine or aspartate aminotransferase levels three times above the upper limit of normal. However, no significant differences regarding the risk of adverse events leading to treatment discontinuation between alirocumab and evolocumab (relative risk = 0.98; 95% CI 0.79–1.22; *p*  =  0.89) were found [83]. Relative to the risk of developing new-onset diabetes, evolocumab and alirocumab were primarily related to mild hyperglycemia rather than diabetes, with most of the hyperglycemic events occurring during the first 6 months of treatment. Thus, careful monitoring and identification of individuals more susceptible to developing diabetes is advised [84].

### 4.5. Drug Interaction

In patients with primary hypercholesterolemia or mixed dyslipidemia receiving a high dose of statin, the systemic exposure to evolocumab was slightly lower than that observed in subjects receiving a low-to-moderate-dose statin. An approximately 20% increase in clearance is partly mediated by the increase in PCSK9 concentration caused by statins [39,85]. This had no negative impact on the pharmacodynamic effect of evolocumab. Alirocumab exposure is approximately 40%, 15%, and 35% lower when used with statins, ezetimibe, and fenofibrate, respectively, compared to when given alone; the cholesterol-lowering effect is not affected [39].

### 4.6. Clinical Efficacy

PCSK9 inhibitors (PCSk9i) are safe and effective in clinical practice, allowing the achievement of the recommended LDL-C target in most patients, especially when used as a combination therapy [86]. These are the conclusions of the AT-TARGET-IT (efficacy, safety, adherence, and persistence of PCSK9 inhibitors in clinical practice) observational Italian registry on the use of PCSK9i in clinical practice. The efficacy and safety of evolocumab and alirocumab have been extensively evaluated in the FOURIER (Further Cardiovascular Outcomes Research with PCSK9 Inhibition in Subjects with Elevated Risk) and ODYSSEY OUTCOMES (Evaluation of Cardiovascular Outcomes After an Acute Coronary Syndrome During Treatment With Alirocumab) studies, respectively, as well as in the subsequent subanalyses [6]. In the FOURIER study enrolling patients with ASCVD, evolocumab reduced MACE by 15%, an effect primarily driven by reduced rates of myocardial infarction (−27%), stroke (−21%), and coronary revascularization (−22%). The benefit was also demonstrated in the open-label extension (FOURIER-OLE), in which 6635 of these patients were transitioned to open-label evolocumab regardless of their initial treatment allocation in the parent trial and were followed up for an additional median of 5 years. Long-term achievement of lower LDL-C levels, down to <20 mg/dL, was associated with a lower risk of cardiovascular outcomes with no significant safety concerns [87].

Numerous secondary analyses of the FOURIER trial showed that patients who experienced a greater absolute risk reduction were those at a higher risk for major vascular events (e.g., those with greater atherosclerotic plaque burden [88], with peripheral artery disease [89], with recent MI [90,91], diabetes [92], metabolic syndrome [93], or prior ischemic stroke [94]). However, although in the FOURIER study the cardiovascular benefits were not different in men vs. women [95], in clinical practice, PCSK9 appear to be less effective in reducing LDL-c levels in women compared to men, implying the importance of sex differences in PCSK9 metabolism [96]. 

Relative to alirocumab, the ODYSSEY OUTCOMES trial showed a reduction in MACE by 15% compared to the placebo in patients with recent acute coronary syndrome. This benefit was evident irrespective of age [97] and independent of baseline eGFR, across a broad range above 30 mL/min/1.73 m^2^, with larger relative risk reductions in patients with eGFR > 60 mL/min/1.73 m^2^ [98]. Other major findings associated with alirocumab were the risk reduction for stroke [99] and total hospitalizations, with a corresponding small rise in days alive and out of hospital [100]. Very recently, data from the ARCHITECT (Effect of Alirocumab on Atherosclerotic Plaque Volume, Architecture and Composition) study showed that alirocumab favored the regression of coronary plaque burden and plaque stabilization (as assessed by coronary computed tomographic angiography) over 78 weeks in patients with familial hypercholesterolemia without clinical atherosclerotic cardiovascular disease. Coronary plaque burden was reduced from 34.6% (32.5–36.8%) to 30.4% (27.4–33.4%), whereas when the characteristics of coronary atherosclerosis were taken into consideration, there was a rise in the proportion of calcified and fibrous plaque, respectively, by +0.3% and +6.2%, a phenomenon accompanied by a decrease in fibro-fatty (3.9%) and necrotic plaque, respectively, by −3.9% and −0.6% [101].

PCSK9 inhibitors lower plasma lipoprotein(a) concentrations by approximately 27%, with no significant difference between evolocumab and alirocumab. PCSK9 inhibitors increase the catabolism of lipoprotein(a) when used as a combination therapy or on a background of statin treatment in high-risk patients. However, as monotherapy, these agents mainly act through the reduction of the lipoprotein(a) production rate [102,103].

## 5. Inclisiran

Small interfering RNA (siRNA) therapy represents the most successful pharmacological approach for gene silencing. Inclisiran is a stabilized double-stranded ribonucleic acid siRNA, conjugated to an N-acetylgalactosamine (GalNAc) and designed to target selectively hepatic PCSK9 mRNA [104,105].

### 5.1. Mechanism of Action

Inclisiran is rapidly and specifically taken up by hepatocytes, ensuring a long-term silencing of PCSK9 expression. This action is associated with a higher LDL receptor recycle and expression on the hepatocyte cell surface, which increases LDL-C uptake and decreases circulating LDL-C levels, resulting in an important cholesterol-lowering effect [105,106]. The mechanism of action of inclisiran is similar to that of all other siRNAs: binding to the RNA-induced silencing complex (RISC) complex composed of siRNAs and proteins involving RNA interference, which promotes the degradation of PCSK9 mRNA by preventing the encoded protein synthesis [105].

### 5.2. Pharmacodynamics

Single s.c. administration of 300 mg of inclisiran resulted in a reduction of PCSK9 levels, compared to the placebo, by 75% and LDL-C by 50% [104]. The effect of inclisiran, both on PCSK9 and on LDL-C, is very significant, not only in terms of efficacy but also for the long duration of action which lasts more than 180 days after a single administration [105]. The drug has recently been approved by both the FDA and European Medicine Agency (EMA) for the treatment of adults with primary hypercholesterolemia (heterozygous familial and non-familial) or mixed dyslipidemia. The recommended dose of inclisiran is 284 mg administered as an initial single s.c. injection followed by a second injection at 3 months, and then every 6 months thereafter. The 284 mg inclisiran dose is equivalent and is referred to as 300 mg inclisiran sodium.

### 5.3. Pharmacokinetics

Following a single subcutaneous administration, at the recommended dose regimen of 284 mg, the C_max_ was reached after approximately 4 h and the drug reached undetectable levels within 48 h after dosing [107]. The pharmacokinetic is linear with increasing doses and following multiple subcutaneous administrations, the parameters are similar to those of single dosing [107]. 

Inclisiran is 87% bound to albumin and the apparent volume of distribution is approximately 500 L. Based on non-clinical data, inclisiran has been shown to have high hepatic extraction, the target organ for cholesterol lowering [107]. Inclisiran is primarily metabolized by nucleases to inactive shorter nucleotides. The terminal elimination half-life (t_1/2_) of inclisiran is approximately 9 h and no accumulation occurs with repeated dosing. Sixteen percent (16%) of inclisiran is eliminated through the kidneys.

Pharmacokinetic analysis, from a dedicated renal impairment study, reported an increase in inclisiran C_max_ of approximately 2.3-, 2.0-, and 3.3-fold and an increase in inclisiran AUC of approximately 1.6-, 1.8-, and 2.3-fold in patients with mild (creatinine clearance [CrCL] 60 mL/min to 89 mL/min), moderate (CrCL 30 mL/min to 59 mL/min), and severe (CrCL 15 mL/min to 29 mL/min), respectively, renal impairment compared to patients with normal renal function. Despite the higher transient plasma exposure of more than 48 h, the LDL-C reduction was similar in all renal function groups. Based on population pharmacodynamic modeling, no dose adjustment is recommended in patients with end-stage renal disease [107]. The effect of hemodialysis on the pharmacokinetics of inclisiran has not been studied. Considering that inclisiran is eliminated by the kidneys, hemodialysis should not be performed for at least 72 h after inclisiran administration.

An inclisiran C_max_ increase of approximately 1.1- and 2.1-fold and AUC of 1.3- and 2-fold has been observed in patients with mild (Child–Pugh class A) and moderate (Child–Pugh class B) hepatic impairment versus patients with normal hepatic function, respectively [108]. Despite the higher transient plasma exposure of inclisiran, reductions in LDL-C were similar between the groups of inclisiran-treated patients with normal hepatic function and mild hepatic impairment. No dose adjustment is necessary in patients with mild and moderate hepatic impairment (Child–Pugh class A and B) [108]. Inclisiran has not been studied in patients with severe hepatic impairment (Child–Pugh class C). Age, body weight, gender, race, and creatinine clearance did not significantly affect the pharmacodynamics of inclisiran [109]. 

### 5.4. Safety

The only adverse reactions associated with inclisiran were injection site reactions (pain, redness, rash), that were mild to moderate and transient. This side effect was more frequent with inclisiran than with the placebo in both ORION-10 and -11 trials, with between-group differences of 1.7 percentage points in the first and 4.2 percentage points in the second [110]. Antidrug antibodies were detected in 2.0% and 2.5% of the samples from inclisiran-treated patients in the ORION-10 and ORION-11 trials, respectively [110]. These patients showed no clinically meaningful differences in the efficacy, safety, or pharmacodynamic profiles of inclisiran.

### 5.5. Drug Interaction

Inclisiran is not a substrate for common drug transporters and is not expected to be a substrate for CYP450. Inclisiran is not an inhibitor or inducer of CYP450 enzymes or common drug transporters. Therefore, no clinically significant interactions are expected with other medicinal products, including statins.

### 5.6. Clinical Efficacy

The efficacy of inclisiran was evaluated in three phase 3 studies, in patients with ASCVD (coronary artery disease, cerebrovascular disease, or peripheral arterial disease), with ASCVD risk equivalents, and/or with familial hypercholesterolemia (FH) [111]. Patients were taking the maximally tolerated dose of a statin with or without other lipid-lowering therapies and required additional LDL-C reduction (patients who were unable to achieve their treatment goal). Patients were administered subcutaneous injections of 284 mg inclisiran or the placebo on day 1, day 90, day 270, and day 450. Patients were followed through until day 540. In the pooled phase 3 analysis, inclisiran administered subcutaneously decreased LDL-C by 50% to 55% as early as day 90 [111]. The effect of inclisiran on cardiovascular morbidity and mortality has not yet been established but studies are ongoing [112,113], including the phase 3 ORION-4 study, which evaluates the effect of inclisiran administration on cardiovascular events in approximately 15,000 patients with atherosclerotic cardiovascular disease, with a planned 5-year follow-up [114]. The efficacy of inclisiran on the global lipid profile observed in the ORION-11 trial is reported in Table 3 [115].

## 6. Bempedoic Acid

Systematic chemical modification of long-chain hydrocarbons has been one approach to identify novel lipid-lowering molecules for the treatment of human dyslipidemias [116]. Over the last 30 years, a variety of classes of compounds have been identified as having lipid-lowering activities in animal models of dyslipidemia. Proposed primary mechanisms of action have included alterations in enzyme activities through allosteric or redox state changes and modulation of gene expression through the activation or inhibition of nuclear hormone receptors. In 2004, a novel ω-hydroxy-alkanedicarboxylic acid, ESP55016, was described that favorably altered serum lipid profiles in an animal model of diabetic dyslipidemia. ESP55016 (8-hydroxy-2,2,14,14-tetramethyl pentadecanedioic acid) was than developed as ET-1002, and named bempedoic acid, which is a first-in-class, oral, small-molecule cholesterol synthesis inhibitor with different mechanism of action than statins [117] (Figure 4).

### 6.1. Mechanism of Action

Bempedoic acid is a direct and competitive inhibitor of adenosine triphosphate citrate lyase (ACLY) [118]. To manifest its direct and competitive inhibitory effect on cholesterol synthesis, bempedoic acid requires its transformation from a prodrug (8-hydroxy-2,2,4,14-tetramethyl-pentadecanedioic acid) to an active metabolite catalyzed by enzyme “very long-chain acyl-CoA synthetase-1 (ACSVL1)” which occurs in hepatocytes and leads to the genesis of the powerful direct and competitive inhibitor, bempedoic acid-CoA [118]. Bempedoic acid is thus activated when acetyl-Coenzyme A is added to the molecule and its reversible conversion to ESP15228, also an active metabolite, is obtained by oxidation of bempedoic acid [119]. ESP15228 contributes minimally to the overall clinical activity of bempedoic acid due to its low systemic exposure and pharmacokinetic properties [118,119].

The ACSVL1 enzyme is selectively expressed in the liver, but not in skeletal muscles, which causes bempedoic acid to be activated only in the liver and have no muscle involvement [118]. This results in the reduction of potential adverse effects on the muscles which, instead, tend to occur during the use of statins.

### 6.2. Pharmacodynamics

As documented by a series of in vitro and in vivo preclinical data, the inhibition of ACLY determines a reduction of cholesterol synthesis in the liver, which translates into an increased expression of the LDL receptor. Thus, bempedoic acid reduces LDL-C (20–25%), non-HDL-C (19%), apo B (15%), and total cholesterol (16%) in patients with hypercholesterolemia or mixed dyslipidemia [120], while no effect is observed on triglycerides levels (+0.4%) [120,121]. Bempedoic acid may help to control cholesterol levels, either as monotherapy or in combination with existing lipid-lowering therapy, in a broad spectrum of patients at high cardiovascular risk [1,122]. 

The selective action of bempedoic acid at the hepatic level on the endogenous synthesis of cholesterol presents a strong rationale, from a pharmacological point of view, for its combination with ezetimibe, an inhibitor of cholesterol intestinal absorption, in the same pharmaceutical formulation [123]. This combination is very effective in the treatment of high- and very high-cardiovascular-risk patients who require significant reductions in LDL-C. In fact, the combination of bempedoic acid, ezetimibe, and statins reduces LDL-C levels [124] by 63%, with an additional 30% with PCSK9 inhibitors [125]. 

### 6.3. Pharmacokinetics

After oral administration, bempedoic acid is absorbed from the small intestine with a T_max_ of 3.5 h (Table 4) [119]. Concomitant administration of food had no effect on its oral bioavailability. The apparent volume of distribution of bempedoic acid is 18 L [119]. The binding of bempedoic acid, its glucuronide, and its active metabolite to plasma proteins is 99.3%, 98.8%, and 99.2%, respectively.

The metabolism of bempedoic acid from prodrug to active drug takes place in hepatocytes, leading to the generation of two active compounds: bempedoic acid with Acyl-CoA and ESP15228, also active metabolite obtained through reversible oxidation by aldo-keto reductase [119]. The major route of elimination of bempedoic acid is via metabolism to acyl glucuronide [119]. At steady state, the mean half-life of bempedoic acid in humans is 19 h (Table 4).

No dose adjustment is necessary in patients with mild/moderate renal impairment. Limited data are available for patients with severe CKD (eGFR < 30 mL/min/1.73 m^2^) and patients with end-stage renal disease on dialysis have not been studied with bempedoic acid [126]. Additional monitoring for adverse reactions when the drug is administered may be appropriate for these patients.

No dose adjustment is necessary in patients with mild or moderate hepatic impairment (Child–Pugh A or Child–Pugh B). Since no data are available in patients with severe hepatic impairment (Child–Pugh C), they should undergo periodic liver function tests. This recommendation is due to the unknown effects of increased exposure to ezetimibe when given in fixed combinations. The safety and efficacy of the drug in children aged less than 18 years have not been established. 

### 6.4. Safety

The drug has a satisfactory safety profile except for an increased risk of gout [127]. Data from phase 3 studies showed an increase in serum uric acid and creatinine levels in patients treated with bempedoic acid [128]. This effect is due to the inhibition of OAT2 by bempedoic acid. Indeed, OAT2 mediates the renal excretion of uric acid. Thus, patients under treatment with bempedoic acid should be monitored for uric acid and creatinine levels. These effects are transient and reversible upon the discontinuation of bempedoic acid treatment. Thus, clinicians are required to assess gout risk factors before committing patients to bempedoic acid, provide dietary guidance, consider alternative antihypertensive medications for at-risk patients currently taking thiazides or loop diuretics, and monitor patients on bempedoic therapy for gout symptoms, promptly initiating management if needed [129]. Relative to new-onset diabetes, data from the CLEAR Outcomes study showed that the proportion of patients who developed new-onset diabetes was similar between the bempedoic acid and placebo groups, with 429 of 3848 (11.1%) with bempedoic acid versus 433 of 3749 (11.5%) with the placebo (HR = 0.95; 95%CI 0.83–1.09) [130]. The same trial highlighted a higher incidence of cholelithiasis (2.2% vs. 1.2%) in the bempedoic acid group compared to the placebo. This should be related to the inhibition of the OATP1B1/3 expressed mainly in hepatocytes [131].

### 6.5. Drug Interactions

Bempedoic acid as well as its active metabolite and its glucuronide are not substrates or inhibitors/inducers of CYP450 involved in drug metabolism [119]. Similarly, they are not substrates of commonly characterized drug transporters, except for bempedoic acid glucuronide, which is a substrate for OAT3 [132]. Bempedoic acid and its glucuronides are weak inhibitors of the transporter proteins organic anion-transporting polypeptide 1B1 (OATP1B1) and 1B3 (OATP1B3) involved in the uptake by hepatocytes of different drugs such as atorvastatin, pravastatin, fluvastatin, pitavastatin, rosuvastatin, and simvastatin [132]. Co-administration with bempedoic acid may lead to increased plasma concentrations of these medicinal products. However, the only clinically relevant interaction of bempedoic acid has been observed with high doses of simvastatin. A single dose of simvastatin 40 mg with bempedoic acid 180 mg at steady state resulted in a 2-fold increase in exposure to simvastatin acid [132]. From this evidence, it is indicated that simvastatin should not exceed 20 mg or 40 mg with bempedoic acid.

### 6.6. Clinical Efficacy

Bempedoic acid has been approved for clinical use following the positive results of four randomized phase 3 studies (CLEAR Harmony, CLEAR Wisdom, CLEAR Tranquility, and CLEAR Serenity) and its clinical efficacy was confirmed in 2023 with the CLEAR Outcomes trial [7,133]. The CLEAR Outcomes study enrolled a total of 13,970 patients who underwent randomization (between December 2016 and August 2019): 6992 were assigned to the bempedoic acid group (180 mg) and 6978 to the placebo group. The median duration of follow-up was 40.6 months. The primary endpoint (death from cardiovascular causes, nonfatal myocardial infarction, nonfatal stroke, or coronary revascularization) occurred in 11.7% of participants taking bempedoic acid and 13.3% of those taking a placebo (HR = 0.87; 95%CI, 0.79–0.96), with a number-needed-to-treat (NNT) of 63. This benefit was in line with the CTTC endpoint calculation, which predicts an HR of 0.846 for 26.1 mg/dL in LDL-C reduction. There was no significant difference in rates of death between the two study arms. The key secondary endpoint (fatal or nonfatal myocardial infarction) occurred in 3.7% of patients given bempedoic acid and in 4.8% of those given a placebo (HR = 0.77; 95%CI, 0.66–0.91). Bempedoic acid was superior to the placebo to reduce coronary revascularization by 19% (HR = 0.81; 95%CI, 0.72–0.92) [7] and also reduced the total number of cardiovascular events [134]. A recent subanalysis showed that in the on-treatment Per Protocol Set (PPS) analysis, LDL-C reduction was 0.71 mmol/L, with an HR for major vascular events of 0.80 (95%CI 0.71–0.89). When normalized to a 38.5 mg/dL reduction in LDL-C, the HR for PPS was even better, 0.73 (0.62–0.85), and falls within the benefit driven by statins in the CTT meta-analyses (Risk Ratio of 0.78 (95%CI 0.76–0.80)) [135]. Overall, bempedoic acid had no clinically important impact on lipoprotein(a) [121].

## 7. Clinical Point of View

Although statins remain the mainstay of LDL-C lowering for CV risk management, it must be recognized that intolerance and aversion to statin therapy is increasing and that the ability to reach the most aggressive LDL-C goals is far from being an easy task, even among takers of high-dose, high-potency statins. Therefore, the availability of several non-statin medications allows the practitioner to tailor therapy to needs and circumstances. The potency of PCSK9-inhibiting agents should be balanced with their cost and invasiveness, and therefore reserved for high-risk patients who need a large additional LDL-C reduction (say, 35–60%) on top of the statin effect (if a statin is tolerated). On the other hand, oral agents with a synergistic mechanism to that of statins (such as ezetimibe and bempedoic acid) should be preferred for patients who need more modest additional LDL-C lowering (say, 20–35%) or those who would not (due to aversion) or cannot (due to cost or insurance coverage) consider injectable therapy. An insistence on statins use for patients who are believed to be intolerant, though justified by cost considerations, is not supported by the current global expert consensus that benefits are driven by LDL-C lowering and not by the type of drug used. Such consensus is so entrenched that efforts to re-purpose drugs like oral CETP inhibitors (originally developed as HDL-C-raising agents) for LDL-C lowering appear likely to succeed [136,137].

## 8. Conclusions

Epidemiological, genetic, and intervention studies have demonstrated that LDL-C is unquestionably causal in the development of atherogenesis and ASCVD. Statins represent the first-line intervention to reduce the atherosclerotic burden driven by raised levels of LDL-C. However, considering the opportunity of reducing LDL-C to very low levels to mitigate the CV risk in very high-risk patients, statin monotherapy is not sufficient in most cases to achieve the required benefit. A combination with ezetimibe, PCSK9 inhibitors (mAbs and siRNA), and the most recent, bempedoic acid, should all be considered as first-line strategy for an optimal LDL-C-lowering action for the prevention of CV disease. Clinical data from all these pharmacological interventions have clearly indicated that, independently from the mechanism of action, LDL-C levels should be lowered as much as possible to prevent CVD, especially in high and very high-risk patients. Nevertheless, many epidemiological data, including those from the Copenhagen General Population Study (CGPS), have indicated that apoB and non-HDL-C can be used to identify patients who remain at high residual risk despite low LDL-C concentrations [138]. Thus, the measurements that include both LDL-C and VLDL-C (such as apoB and non-HDL-C) should be more reflective of overall ASCVD risk and new therapies should implement the pharmacological control of these lipoproteins. Antithrombotic and anti-inflammatory therapies also need to be developed to reduce the cardiovascular risk. The recent FDA approval of colchicine for decreasing CV events could have a positive impact for a combination therapy aimed at controlling dyslipidemia and chronic inflammation [139,140]. In addition, preclinical and clinical evidence has indicated that statins [141,142] and, more recently, mAbs anti-PCSK9 [143] may elicit some pleiotropic effects beyond lipid-lowering activities, which may contribute to the final observed positive clinical outcomes. 

## Figures and Tables

**Figure 1 jcm-13-00943-f001:**
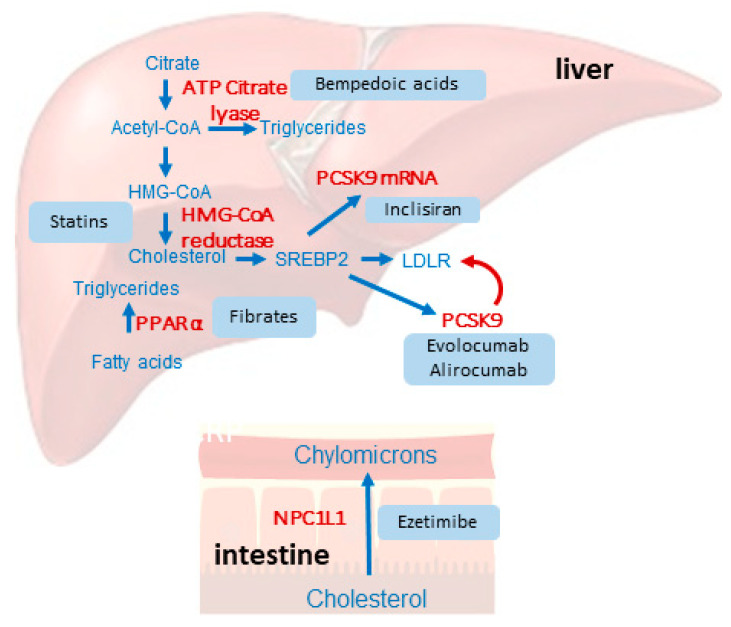
Schematic overview of the different hypolipidemic drugs and their mechanism of action.

**Figure 2 jcm-13-00943-f002:**
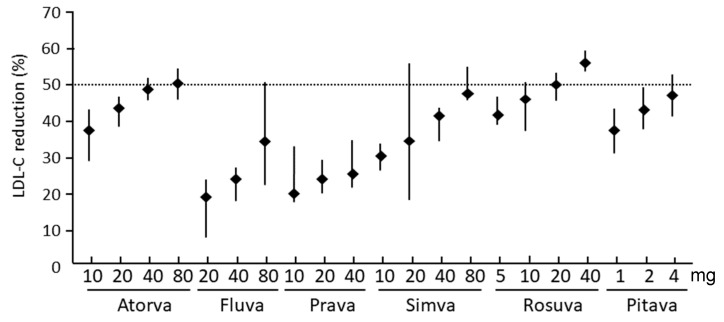
Dose-dependent lipid-lowering potency of statins.

**Figure 3 jcm-13-00943-f003:**
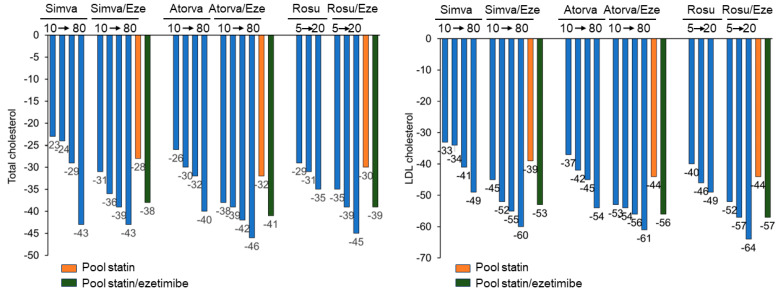
Lipid-lowering effect of ezetimibe in combination with statins.

**Figure 4 jcm-13-00943-f004:**
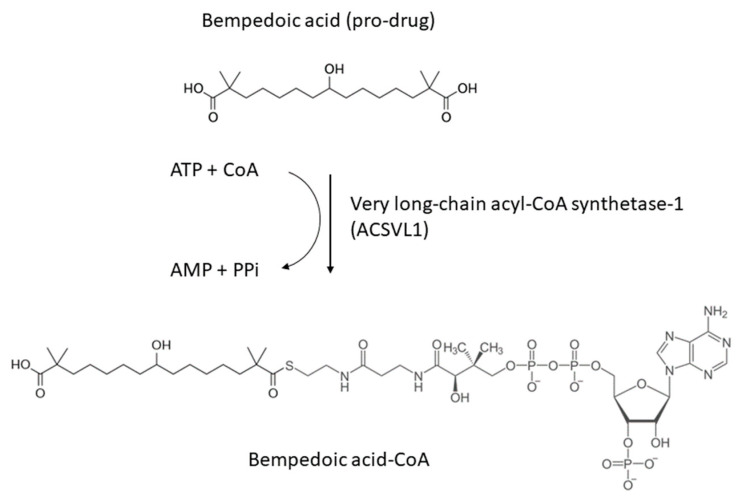
Chemical structure of bempedoic acid and its active metabolite.

**Table 1 jcm-13-00943-t001:** Pharmacokinetic properties of statins.

Parameter	Lova	Simva	Prava	Fluva	Atorva	Rosuva	Pitava
Solubility	Lipophilic	Lipophilic	Hydrophilic	Lipophilic	Lipophilic	Hydrophilic	Lipophilic
Oral absorption (%)	30	60–85	35	98	30	50	80
Protein binding (%)	>95	>95	50	>98	>98	88	96
Bioavailability (%)	<5	<5	18	30	12	20	51
Hepatic extraction (%)	83	≥80	45	≥70	70	63	Estimated > 70
Half-life (h)	1–3	2–5	1–3	1–3	15–30	20	10–13
Renal excretion (%)	10	13	20	6	<5	10	<4
CYP450 metabolism	3A4	3A4	Sulfonation	2C9, 2C8/3A4 (minor)	3A4	Biliary, 2C9, 2C19 (minor)	Biliary, 2C9, 2C8 (minor)
UGTA1/1A3 metabolism	+	+	+	+	+	+	+
P-gp substrate	+	+	+	+	+	+	+
MRP2	+	+	+	+	+	+	+
OATP1B1	+	+	+	+	+	+	+
OATP2B1	N/A	N/A	+	+	+	+	+
BCRP	+	+	+	+	+	+	+

CYP, cytochrome; BCRP, breast cancer resistance protein; MRP2, multidrug resistance protein 2; OATP, organic anion transporting polypeptides; UGT, uridine-diphospho glucuronosyl transferase; N/A, not available. Reproduced with permission from Ruscica, M. et al. [11]. Atorva: atorvastatin; Fluva: fluvastatin; Lova, lovastatin; Prava: pravastatin; Simva: simvastatin; Rosuva: rosuvastatin; Pitava: pitavastatin.

**Table 2 jcm-13-00943-t002:** Pharmacokinetics and pharmacodynamics properties of monoclonal antibodies anti-PCSK9.

	Evolocumab and Alirocumab
Characteristic	Monoclonal antibodies
Mechanism of action	Inhibits PCSK9 binding to the LDLR
Site of action	Circulating PCSK9 (hepatic and extrahepatic origin)
Onset of action on PCSK9	4 h max inhibition
Onset of action on LDL-C	6 days—50% LDL-C
Free PCSK9 level	Decreased (−97%)
PCSK9 level	Increased
Injection frequency	2–4 weeks
CKD	Pharmacokinetics not affected
Bioavailability	70%
T_max_	3–4 days
Vd	3.4 L
T_1/2_	11–17 days

CKD, chronic kidney disease; LDL-C, low-density lipoprotein cholesterol; PCSK9, proprotein convertase subtilisin/kexin type 9; T_1/2_, time it takes for the plasma concentration of a drug or the amount of drug in the body to be reduced by 50%; Vd, volume of distribution.

**Table 3 jcm-13-00943-t003:** Lipid-lowering effect of inclisiran. Modified from Ray, K.K et al. [115].

Parameter	Inclisiran	Placebo	Placebo Adjusted	*p* Value
Total cholesterol	−24.7%	+0.3%	−25.1%	<0.0001
LDL-C	−40.4%	+0.6%	−41.0%	<0.0001
ApoB	−35.5%	−0.7%	−34.8%	<0.0001
Non-HDL-C	−35.5%	−0.2%	−35.3%	<0.0001
Triglycerides	−5.9%	2.5%	−8.4%	=0.045
Lipoprotein(a)	−12.1%	+16.8%	−28.9%	<0.0001
HDL-C	+11%	+6.0%	+5.0%	=0.02

ApoB, apolipoprotein B; HDL-C, high-density lipoprotein cholesterol; LDL-C, low-density lipoprotein cholesterol.

**Table 4 jcm-13-00943-t004:** Pharmacokinetics characteristics of bempedoic acid.

Posology	Oral 180 mg Daily
Effect of food on absorption	None
T_max_ (h)	3.5
Vd (L)	18
Protein bound	99%
Prodrug	Yes
Active metabolites	Yes
Metabolism	Glucuronidation UGT2B7
Substrate of transporters	OAT1B1/3, OAT2, OAT3
Excretion	Prodrug: 5% urine and feces Active metabolites and conjugated: 70% kidney; 30% liver
Half-life (h)	15–24
Clearance (mL/min)	11.2

OAT, organic anion transporting; UGT, UDP-Glycosyltransferase; Vd, volume of distribution; T_max_, time of peak plasma concentration.

## Data Availability

Not applicable.

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
