# Peer review of "Low-Density Lipoprotein Cholesterol-Lowering Drugs: A Narrative Review"

_jcm, 2024, doi:10.3390/jcm13040943_

Round 1
Reviewer 1 Report
Comments and Suggestions for Authors
This article provides a comprehensive and timely review of LDL cholesterol lowering drugs in a well-organised, clear and consistent format which make for an informative read.
Suggested minor points to correct:
- Clarify the confusing wording (lines 194-196) "the history of ezetimibe..."
- Proof read for typographical errors for example: Line 14 "inhibits", Line 72 "Pharmacodynamics", Line 93 add a comma between "atorvastatin, rosuvastatin", Line 115 change "been never" to "never been", Line 128 delete "to" from "undergo to drug", Line 164 change "doubling" to "double" etc.
- Double-check the reference for lines 104-106, As reference 17 seems solely focused on rosuvastatin and not Atorvastatin.
- Clarify of it was "over 6000 bacterial strains" or 6000 microbial strains (bacteria and fungal strains were investigated?) (line 55).
- Ensure all acronyms are defined ("PK" line 168), ("RR" line 182), (MI, line 187), (MACE, line 192),
- Figure 2: Graph is missing units of measurement on the X axis.
- Double-check if lovastatin might still be available in the US as well as these 6 statins mentioned in line 60.
- Provide reference for Line 123-125, as far as I can see, ref 21 does not include information regarding statin liver toxicity so I think there may be a reference missing.
- Confirm reference 31 (line 165) is correct.
- Line 216 confirm if this should be 2 or 4 weeks - Although the study did run in full for 4 weeks, it was split up into two groups of two weeks of treatment with a 2 week break in the middle and the results reported on in this review from this study, are noted for after only 2 weeks of study (based on table 2 and 3 of the cited article).
- Confirm if "non-fat" should be changed to "fasted".
- Line 240, confirm is reference 44 if correct and includes radio-labelled ezetimibe.
- Line 258 "10,000 patients" This is probably true, but the only study that is mentioned in the rest of this paragraph (ref 49), had 3030 patients, and so it seems potentially misleading to mention more than 10,000 here when none of these studies are referenced.
- Line 271: Maybe just briefly add what closporine is in brackets (e.g. an immunosuppresent), to add some context for the readers.
- Confirm reference 110 is correct for statements made lines 548-552.
Points I feel should be clarified:
- How is "complete statin intolerance" defined? (line 116)
- Why might this figure be overestimated? Maybe provide some evidence for this? (line 117)
- Clarify confusing wording (line 121), at first I thought that it might be a 30% increased risk of developing a further CVD event. But after checking the reference (20) provided, this paper is regarding discontinuation of statin therapy due to muscular side effects. Noting that statin induced muscular symptoms were experienced by 10% of statin users, which led to treatment discontinuation of 30% of symptomatic patients.
- Note that reference 26 (line 152 and 154), specifically notes that lovastatin and simvastatin are NOT associated with an increased risk of stroke but they are associated with increased risk of hemorrhage events. It is important to note that the hemorrhage events encompass upper and lower gastrointestinal events and also “other” hemorrhage events, and so I don’t think it is possible to specifically say that these statins increase the risk of hemorrhagic stroke from this reference.
- Line 183-185: I think it might be important to note that this is from one study conducted in the US and so not sure how it necessarily reflects in terms of a global perspective. Maybe specify that this figure is only from one study in one country?
- Line 599 "epidemic proportions" - this wording feels very strong and yet vague. Could the authors define clearly what this means and provide reference? - this is not covered explicitly in sections 2.4 or 2.6 so also may be worth including here too.
Suggested points I feel would add further value to the manuscript if the authors were to incorporate:
- Defining what "very low levels" of LDL-C are (line 39).
- Define the generations that they have set the statins into, as I have seen varying generation definitions used in the literature previously, and also varying opinions as to which statins are in which generation (line 57).
- Add a sentence as to why/ how statins mostly target hepatocytes (line 68).
- Provide a percentage/estimate of HDL-C as well, as they have provided for all the other changes mentioned in this sentence and the one before (line 75).
- As lovastatin and cerivastatin are also mentioned throughout the statin section of this review, may also be useful to include them in Table 1?- maybe highlight that cerivastatin has been discontinued if added to table.
- Including a summary figure that depicts the various different mechanisms of action of the different drugs.
Comments on the Quality of English Language
Minor editing required
Author Response
- Clarify the confusing wording (lines 194-196) "the history of ezetimibe..."
Response: we have now simplified this part of the manuscript that now reads: “Ezetimibe was first designed as potential ACAT (acylcoenzyme A cholesterol acyltransferase) inhibitor and subsequently was discovered to reduce cholesterol intestinal absorption. Thus, ezetimibe was identified without a clear understanding of its molecular target which was also unknown at the time. Thus, ezetimibe is a selective inhibitor of intestinal cholesterol absorption.
- Proof read for typographical errors for example: Line 14 "inhibits", Line 72 "Pharmacodynamics", Line 93 add a comma between "atorvastatin, rosuvastatin", Line 115 change "been never" to "never been", Line 128 delete "to" from "undergo to drug", Line 164 change "doubling" to "double" etc.
Response: we thank the reviewer for these comments. We have now corrected these errors and checked for others throughout the manuscript.
- Double-check the reference for lines 104-106, As reference 17 seems solely focused on rosuvastatin and not Atorvastatin.
Response: we double check the appropriateness of the reference and confirmed that in the following manuscript table 3 described the pharmacokinetics characteristics of all statins. Ruscica, M.; Ferri, N.; Banach, M.; Sirtori, C.R.; Corsini, A. Side effects of statins: from pathophysiology and epidemiology to diagnostic and therapeutic implications. Cardiovascular research 2023, 118, 3288-3304, doi:10.1093/cvr/cvac020
- Clarify of it was "over 6000 bacterial strains" or 6000 microbial strains (bacteria and fungal strains were investigated?) (line 55).
Response: we thank the reviewer for the comment. it is indeed microbial and not bacterial. We have now change this sentence accordingly.
- Ensure all acronyms are defined ("PK" line 168), ("RR" line 182), (MI, line 187), (MACE, line 192),
Response: we deleted PK and used pharmacokinetics. We have indicated RR as relative risk, M as myocardial infarction and MACE as major adverse cardiovascular events.
- Figure 2: Graph is missing units of measurement on the X axis.
Response: We thank the reviewer. We have indicated mg as unit of X axis. After revision the figure 2 became figure 1.
- Double-check if lovastatin might still be available in the US as well as these 6 statins mentioned in line 60.
Response: We agree with the reviewer. We have now added lovastatin in Table 1.
- Provide reference for Line 123-125, as far as I can see, ref 21 does not include information regarding statin liver toxicity so I think there may be a reference missing.
Response: We have now checked and indicated references 18. This sentence now reads: Statins are safe and well-tolerated although some patients may develop muscular complications, type 2 diabetes, and increased levels of liver aminotransferases 18.
18) Ruscica, M.; Ferri, N.; Banach, M.; Sirtori, C.R.; Corsini, A. Side effects of statins: from pathophysiology and epidemiology to diagnostic and therapeutic implications. Cardiovascular research 2023, 118, 3288-3304, doi:10.1093/cvr/cvac020.
- Confirm reference 31 (line 165) is correct.
Response: we double checked the reference. We confirm that it is correct since describes the DDI between statins and sacubitril. Thus, we confirm the following sentence and the references below. “Finally, the combination of a neprilysin inhibitor, sacubitril, with the angiotensin receptor blocker, valsartan has been shown to double atorvastatin Cmax and AUC, although no clinically significant adverse effects were reported [45].
45) Lin, W.; Ji, T.; Einolf, H.; Ayalasomayajula, S.; Lin, T.H.; Hanna, I.; Heimbach, T.; Breen, C.; Jarugula, V.; He, H. Evaluation of Drug-Drug Interaction Potential Between Sacubitril/Valsartan (LCZ696) and Statins Using a Physiologically Based Pharma-cokinetic Model. Journal of pharmaceutical sciences 2017, 106, 1439-1451, doi:10.1016/j.xphs.2017.01.007.
- Line 216 confirm if this should be 2 or 4 weeks - Although the study did run in full for 4 weeks, it was split up into two groups of two weeks of treatment with a 2 week break in the middle and the results reported on in this review from this study, are noted for after only 2 weeks of study (based on table 2 and 3 of the cited article).
Response: We thank the reviewer for the comment. We changed 4 weeks with 2 weeks. This was, indeed, a crossover study.
54) Sudhop, T.; Lutjohann, D.; Kodal, A.; Igel, M.; Tribble, D.L.; Shah, S.; Perevozskaya, I.; von Bergmann, K. Inhibition of intestinal cholesterol absorption by ezetimibe in humans. Circulation 2002, 106, 1943-1948, doi:10.1161/01.cir.0000034044.95911.dc.
- Confirm if "non-fat" should be changed to "fasted".
Response: We agree with the reviewer and changed the sentence to “The oral bioavailability of ezetimibe was similar under both fed and fasting conditions, and thus the drug can be administered with or without meals [59].
- Line 240, confirm is reference 44 if correct and includes radio-labelled ezetimibe.
Response: We thank the reviewer. We have now indicated the right reference. “Following oral administration of 14C-ezetimibe (20 mg) to humans, approximately 93% of the total radioactivity reached the circulation, and approximately 78% and 11% of the administered radioactivity was recovered in the feces and urine, respectively [62].
62) Patrick, J.E.; Kosoglou, T.; Stauber, K.L.; Alton, K.B.; Maxwell, S.E.; Zhu, Y.; Statkevich, P.; Iannucci, R.; Chowdhury, S.; Affrime, M.; et al. Disposition of the selective cholesterol absorption inhibitor ezetimibe in healthy male subjects. Drug metabolism and disposition: the biological fate of chemicals 2002, 30, 430-437, doi:10.1124/dmd.30.4.430.
- Line 258 "10,000 patients" This is probably true, but the only study that is mentioned in the rest of this paragraph (ref 49), had 3030 patients, and so it seems potentially misleading to mention more than 10,000 here when none of these studies are referenced.
Response: We agree with the reviewer. We have now changed the sentence to: “The tolerability of ezetimibe therapy has been evaluated in several clinical studies”
- Line 271: Maybe just briefly add what closporine is in brackets (e.g. an immunosuppresent), to add some context for the readers.
Response: We agree with the reviewer and change the sentence to: “The concomitant use of ezetimibe and the immunosuppressant cyclosporine A in transplant subjects should be closely monitored as the concentration of ezetimibe was shown to be increased by 12 fold.
- Confirm reference 110 is correct for statements made lines 548-552.
Response. The statement: “No dose adjustment is necessary in patients with mild hepatic impairment (Child-Pugh A) and its use is not recommended in patients with moderate (Child-Pugh B) or severe (Child-Pugh C) hepatic impairment” has been taken from the SUMMARY OF PRODUCT CHARACTERISTICS of bempedoic acid approved in Europe. We cited this document from the EMA website. “Nilemdo: SUMMARY OF PRODUCT CHARACTERISTICS. https://www.ema.europa.eu/en/documents/product-information/nilemdo-epar-product-information_en.pdf 2023”
Points I feel should be clarified:
- How is "complete statin intolerance" defined? (line 116)
Response: we have now extensively developed this part of the manuscript as following” In particular, statin-associated muscle symptoms (SAMS) represent the most common symptoms that result in statin intolerance or statin discontinuation, increasing the risk for recurrent myocardial infarction (50%) and coronary heart disease events (51%) [24]. Specifically, statin intolerance is defined as one or more adverse effects associated with statin therapy, which resolves or improves with dose reduction or discontinuation. It can be classified as complete inability to tolerate any dose of a statin, or partial intolerance, with inability to tolerate the dose necessary to achieve the patient-specific therapeutic objective. To classify a patient as having statin intolerance, a minimum of two statins should have been attempted, including at least one at the lowest approved daily dosage [25]. However, data extrapolated from randomized clinical trials and re-al-world studies are contrasting on this issue (i.e., in long-term clinical trials, muscle adverse events occur at similar rates, regardless of allocation to statin or placebo). In-deed, trials with statins excluded many patients who would be prescribed a statin in clinical practice. These trials excluded patients unlikely to adhere to the demands of a long-term study and those taking known interacting drugs [26]. On this matter, an analysis of 19 placebo-controlled trials concluded that >90% of all reports of muscle symptoms by participants allocated statin therapy were not due to the statin [27]. Conversely, in routine care settings, this percentage can increase until 30% [28,29]. However, rechallenge is possible, by changing statin type and reducing the dose. As established by both interventional studies and clinical practice, it is recommended that before rechallenging, a washout period may be useful to allow symptom resolution, which varies among individuals [30]. A further approach can be the alternate-day dos-ing by prescribing atorvastatin and rosuvastatin which possess a long half-life [31].
- Why might this figure be overestimated? Maybe provide some evidence for this? (line 117)
Response. we thank the reviewer for the suggestion. We have now added the following sentence: “The excess rate of SAMS mainly derives from the so-called nocebo effect, as demon-strated in the ASCOT-LLA study [35]. Indeed, muscle symptoms are referred more often when patients and their doctors are aware that statin therapy has been used and not when its use was blinded.
35) Gupta, A.; Thompson, D.; Whitehouse, A.; Collier, T.; Dahlof, B.; Poulter, N.; Collins, R.; Sever, P.; Investigators, A. Adverse events associated with unblinded, but not with blinded, statin therapy in the Anglo-Scandinavian Cardiac Outcomes Trial-Lipid-Lowering Arm (ASCOT-LLA): a randomised double-blind placebo-controlled trial and its non-randomised non-blind extension phase. Lancet 2017, 389, 2473-2481, doi:10.1016/S0140-6736(17)31075-9.
- Clarify confusing wording (line 121), at first I thought that it might be a 30% increased risk of developing a further CVD event. But after checking the reference (20) provided, this paper is regarding discontinuation of statin therapy due to muscular side effects. Noting that statin induced muscular symptoms were experienced by 10% of statin users, which led to treatment discontinuation of 30% of symptomatic patients.
Response. we agree with the reviewer. we have now extensively changed this part of the manuscript as following: “[24]. Specifically, statin intolerance is defined as one or more adverse effects associated with statin therapy, which resolves or improves with dose reduction or discontinuation. It can be classified as complete inability to tolerate any dose of a statin, or partial intol-erance, with inability to tolerate the dose necessary to achieve the patient-specific therapeutic objective. To classify a patient as having statin intolerance, a minimum of two statins should have been attempted, including at least one at the lowest approved daily dosage [25]. However, data extrapolated from randomized clinical trials and re-al-world studies are contrasting on this issue (i.e., in long-term clinical trials, muscle adverse events occur at similar rates, regardless of allocation to statin or placebo). In-deed, trials with statins excluded many patients who would be prescribed a statin in clinical practice. These trials excluded patients unlikely to adhere to the demands of a long-term study and those taking known interacting drugs [26]. On this matter, an analysis of 19 placebo-controlled trials concluded that >90% of all reports of muscle symptoms by participants allocated statin therapy were not due to the statin [27]. Conversely, in routine care settings, this percentage can increase until 30% [28,29]. However, rechallenge is possible, by changing statin type and reducing the dose. As established by both interventional studies and clinical practice, it is recommended that before rechallenging, a washout period may be useful to allow symptom resolution, which varies among individuals [30]. A further approach can be the alternate-day dos-ing by prescribing atorvastatin and rosuvastatin which possess a long half-life [31].
- Serban, M.C.; Colantonio, L.D.; Manthripragada, A.D.; Monda, K.L.; Bittner, V.A.; Banach, M.; Chen, L.; Huang, L.; Dent, R.; Kent, S.T.; et al. Statin Intolerance and Risk of Coronary Heart Events and All-Cause Mortality Following Myocardial Infarction. J Am Coll Cardiol 2017, 69, 1386-1395, doi:10.1016/j.jacc.2016.12.036.
- Cheeley, M.K.; Saseen, J.J.; Agarwala, A.; Ravilla, S.; Ciffone, N.; Jacobson, T.A.; Dixon, D.L.; Maki, K.C. NLA scientific statement on statin intolerance: a new definition and key considerations for ASCVD risk reduction in the statin intolerant patient. Journal of clinical lipidology 2022, 16, 361-375, doi:10.1016/j.jacl.2022.05.068.
- Newman, C.B.; Tobert, J.A. Statin intolerance: reconciling clinical trials and clinical experience. JAMA 2015, 313, 1011-1012, doi:10.1001/jama.2015.1335.
- Cholesterol Treatment Trialists, C. Effect of statin therapy on muscle symptoms: an individual participant data meta-analysis of large-scale, randomised, double-blind trials. Lancet 2022, 400, 832-845, doi:10.1016/S0140-6736(22)01545-8.
- Zhang, H.; Plutzky, J.; Skentzos, S.; Morrison, F.; Mar, P.; Shubina, M.; Turchin, A. Discontinuation of statins in routine care settings: a cohort study. Annals of internal medicine 2013, 158, 526-534, doi:10.7326/0003-4819-158-7-201304020-00004.
- Casula, M.; Gazzotti, M.; Bonaiti, F.; E, O.I.; Arca, M.; Averna, M.; Zambon, A.; Catapano, A.L.; Group, P.S. Reported muscle symptoms during statin treatment amongst Italian dyslipidaemic patients in the real-life setting: the PROSISA Study. Journal of internal medicine 2021, 290, 116-128, doi:10.1111/joim.13219.
- Rosenson, R.S.; Baker, S.; Banach, M.; Borow, K.M.; Braun, L.T.; Bruckert, E.; Brunham, L.R.; Catapano, A.L.; Elam, M.B.; Mancini, G.B.J.; et al. Optimizing Cholesterol Treatment in Patients With Muscle Complaints. J Am Coll Cardiol 2017, 70, 1290-1301, doi:10.1016/j.jacc.2017.07.752.
- Awad, K.; Mikhailidis, D.P.; Toth, P.P.; Jones, S.R.; Moriarty, P.; Lip, G.Y.H.; Muntner, P.; Catapano, A.L.; Pencina, M.J.; Rosenson, R.S.; et al. Efficacy and Safety of Alternate-Day Versus Daily Dosing of Statins: a Systematic Review and Meta-Analysis. Cardiovasc Drugs Ther 2017, 31, 419-431, doi:10.1007/s10557-017-6743-0.
- Note that reference 26 (line 152 and 154), specifically notes that lovastatin and simvastatin are NOT associated with an increased risk of stroke but they are associated with increased risk of hemorrhage events. It is important to note that the hemorrhage events encompass upper and lower gastrointestinal events and also “other” hemorrhage events, and so I don’t think it is possible to specifically say that these statins increase the risk of hemorrhagic stroke from this reference.
Response: we agree with the reviewer. we have now changed the sentence to: “In patients with non-valvular atrial fibrillation in therapy with dabigatran, a direct oral anticoagulant, the co-administration of simvastatin and lovastatin has been associated with a higher risk of major hemorrhage relative to other statins
- Line 183-185: I think it might be important to note that this is from one study conducted in the US and so not sure how it necessarily reflects in terms of a global perspective. Maybe specify that this figure is only from one study in one country?
Response: we agree with the reviewer. this sentence now reads: “Despite guideline recommendations, among patients with established ASCVD, in United States fewer than one-half are statin users and only 22.5% are on a high-intensity formulation”
- Line 599 "epidemic proportions" - this wording feels very strong and yet vague. Could the authors define clearly what this means and provide reference? - this is not covered explicitly in sections 2.4 or 2.6 so also may be worth including here too.
Response: we agree with the reviewer. we have now changed the sentence as following: “Although statins remain the mainstay of LDL-C lowering for CV risk management, it must be recognized that intolerance and aversion to statin therapy is increasing and that the ability to get to the most aggressive LDL-C goals is far from being an easy task even among takers of high-dose, high-potency statins.
Suggested points I feel would add further value to the manuscript if the authors were to incorporate:
- Defining what "very low levels" of LDL-C are (line 39).
Response: However, since the most recent guidelines recommend reducing LDL-C to levels below 55 mg/dL (1.4 mmol/L) in very high-risk patients, and even lower (<1.0 mmol/L; <40 mg/dL) for patients with ASCVD who experience a second vascular event within 2 years to mitigate the ASCVD risk [3], statin monotherapy is not sufficient in many patients.
3) Mach, F.; Baigent, C.; Catapano, A.L.; Koskinas, K.C.; Casula, M.; Badimon, L.; Chapman, M.J.; De Backer, G.G.; Delgado, V.; Ference, B.A.; et al. 2019 ESC/EAS Guidelines for the management of dyslipidaemias: lipid modification to reduce cardiovascular risk. Eur Heart J 2020, 41, 111-188, doi:10.1093/eurheartj/ehz455.
- Define the generations that they have set the statins into, as I have seen varying generation definitions used in the literature previously, and also varying opinions as to which statins are in which generation (line 57).
Response: we apologize with the reviewer. we have now changed the sentence to: “First generation of statins (lovastatin, simvastatin, and pravastatin) were, therefore, obtained by fermentation, while fluvastatin, defined as second generation, is a synthetic product. Aatorvastatin, rosuvastatin and cerivastatin are the third second generation of statins, again obtained by synthesis”
- Add a sentence as to why/ how statins mostly target hepatocytes (line 68).
Response: We thank the reviewer for the comment. We have now indicated why statins mostly target hepatocytes: The mechanism of action, common to all statins, is based on competitive inhibition of the enzyme HMG-CoA reductase, rate-limiting step of cholesterol synthesis in all cells, but mostly targeting the hepatocytes, due to their high hepatic extraction [8, 11].
8) Ferri, N.; Corsini, A. Clinical Pharmacology of Statins: an Update. Current atherosclerosis reports 2020, 22, 26, doi:10.1007/s11883-020-00844-w.
11) Ruscica, M.; Ferri, N.; Banach, M.; Sirtori, C.R.; Corsini, A. Side effects of statins: from pathophysiology and epidemiology to diagnostic and therapeutic implications. Cardiovascular research 2023, 118, 3288-3304, doi:10.1093/cvr/cvac020.
- Provide a percentage/estimate of HDL-C as well, as they have provided for all the other changes mentioned in this sentence and the one before (line 75).
Response: We thank the reviewer. we have resentence this part as follow: “These drugs slightly reduce triglyceride levels (10-30%) but only in hypertriglyceridemic subjects (i.e., the greater the LDL-C reducing efficacy is, the more effective the triglycerides reduction is) [15]. In addition to these actions, statins reduce ApoB, non-HDL cholesterol, VLDL cholesterol, VLDL triglycerides, and increase ApoA-I and HDL cholesterol (5-10%) [3].
15) Stein, E.A.; Lane, M.; Laskarzewski, P. Comparison of statins in hypertriglyceridemia. Am J Cardiol 1998, 81, 66B-69B, doi:10.1016/s0002-9149(98)00041-1.
3) Mach, F.; Baigent, C.; Catapano, A.L.; Koskinas, K.C.; Casula, M.; Badimon, L.; Chapman, M.J.; De Backer, G.G.; Delgado, V.; Ference, B.A.; et al. 2019 ESC/EAS Guidelines for the management of dyslipidaemias: lipid modification to reduce cardiovascular risk. Eur Heart J 2020, 41, 111-188, doi:10.1093/eurheartj/ehz455.
- As lovastatin and cerivastatin are also mentioned throughout the statin section of this review, may also be useful to include them in Table 1?- maybe highlight that cerivastatin has been discontinued if added to table.
Response: We add lovastatin in table 1 while we preferred to not mention cerivastatin since it is out of market from many years. We hope the reviewer will find our decision reasonable.
- Including a summary figure that depicts the various mechanisms of action of the different drugs.
Response: We have now added a new figure showing the different mechanism of action of the drugs.
Reviewer 2 Report
Comments and Suggestions for Authors
General Comments: I think the major problem is that it isn’t clear who this review is targeting. If it is targeting clinicians there are numerous deficiencies that are noted in my comments to authors. If is targeting pharmacologists there are also few areas that are not addressed. The authors tried to cover an enormous amount of information and therefore doesn’t provide sufficient information particularly as it relates to health care providers. The information presented seems most appropriate for pharmacologists. Perhaps if they were more focused the review would have been more valuable.
Specific Comments
1) “Currently, there are six statins available for clinical use: simvastatin, fluvastatin, atorvastatin, pravastatin, rosuvastatin, and pitavastatin”. There are actually seven statins available. Lovastatin is still available in the US.
2) “These drugs slightly reduce triglycerides levels (10-30%)”. It should be noted that the effect of statins on TG levels is dependent on the patients TG level. The decrease in TGs occurs if TG levels are elevated. Additionally, the reduction in TGs is related to the decrease in LDL-C levels. The greater the reduction in LDL-C the greater the decrease in TGs (PMID: 9526817). This information is important for clinicians.
3) One should mention the effect of statins on Lp(a) levels. As you are aware there is an increasing interest in Lp(a) as a causative factor in atherosclerosis.
4) In discussing the safety of statins ones needs a more extensive discussion. It is crucial that health care providers understand the details regarding the potential side effects of statins.
Muscle- A recent publication by the Cholesterol Treatment Trialists has presented detailed information on muscle complications based on a meta-analysis of 23 trials with over 100,000 participants (PMID: 36049498). Additionally, a number of studies have reported double blinded statin challenge studies in patients with “statin induced” muscle symptoms who were rechallenged with a statin vs. placebo (PMID: 25545331, 27039291). These and other studies should be discussed to put into perspective the effect of statins on muscle. It is essential that clinicians understand the relationship.
Diabetes- The frequency of this side effect should be mentioned and the risk factors for developing this complication discussed (PMID 20167359, 23219296).
Liver- The authors should point out that the difference in the risk of developing elevated transaminases in statin treated and controls is very small (1.4% in statin treated vs. 1.1% in controls; PMID 17159064) and that the FDA no longer recommends following liver function tests in patients treated with statins.
5) If this review is directed at pharmacologists one needs to discuss why gemfibrozil effects statin metabolism (Gemfibrozil inhibits the glucuronidation of statins, PMID: 24793440).
6) If this review is directed at clinicians one should focus on drugs that require an adjustment in the statin dose or make the use of statins contraindicated in the drug interaction section.
7) In the drug interaction section why no mention of amiodarone, diltiazem, and verapamil? These are commonly used drugs. Also, why no mention of grapefruit juice? Our patients often ask about this.
8) Interestingly, in the clinical efficacy section there is no mention of the populations where statins have failed to show clinical benefit. Specifically, in patients with renal disease on maintenance dialysis or congestive heart failure several studies have not shown benefits from statin treatment. It is essential that clinicians know when a drug will not be beneficial.
9) In the ezetimibe section it is stated “The drug has shown, in combination with simvastatin, to reduce the risk of cardiovascular events in relation to the reduction of LDL-C both in nephropathic and in acute coronary syndrome patients”. It should be noted that there is a monotherapy trial where ezetimibe alone decreased cardiovascular events (PMID: 31434507).
10) The authors should emphasize that the combination of low dose statin + ezetimibe lowers LDL-C levels to almost the same extent as high dose statins (see table 2 of your paper). It is important for clinicians to know that one can, in certain patients, use a low dose of a statin with ezetimibe and achieve very good LDL-C reductions similar to intensive statin therapy.
11) The authors note that PCSK9 inhibitors decrease Lp(a) levels. The authors need to mention potential mechanisms for this observation as statins and ezetimibe, which also increase hepatic LDL receptors do not have this effect.
12) In the PCSK9 side effect section the following is stated “The most common adverse reactions reported in phase 3 clinical trials were muscle pain (27.2%), nasopharyngitis (9.3%), back pain (12%), upper respiratory tract infections (9%), headache (9.2%), joint pain (7%), flu-like symptoms (7.5%), or an increase in alanine or aspartate aminotransferase levels three times above the upper limit of normal (6%)”. Are these any different than what is observed in the control group? In studies one often sees common abnormalities, such as headaches, in a similar percentage of participants in the drug group and the placebo group.
13) In the PCSK9 side effect section the following is stated “Most importantly, the PCSK9 inhibition by evolocumab and alirocumab is not associated to higher risk of new cases of diabetes”. It should be noted that both the FOURIER and ODISSEY OUTCOMES were both relatively short trials and therefore could have missed an increase in diabetes. Note it took many years and many trials before the link between statins and diabetes were recognized and individual trials often did not show an increase. If you examine figure 2 in the classic meta-analysis of statin induced diabetes (PMID: 20167359) you will see that the only study that showed a statistically significant increased risk of diabetes was the JUPITER Trial and several actually showed a reduced risk (non-significant). It should be noted that a Mendelian randomization study of PCSK9 variants has raised the possibility of a link with diabetes (PMID: 27908689). Thus, I think further studies are needed. I think it is essential to carefully review the literature before making definitive statements that are clinically very important.
14) It is stated in the PCSK9 efficacy section “in clinical practice, PCSK9 appear to be less effective in reducing LDL-c levels in women compared to men, implying the importance of sex differences in PCSK9 metabolism”. This is based on a real-world study which has major limitations. Are there data from the numerous randomized controlled trials showing a difference in LDL-C response in men and women?
15) The effect of bempedoic acid on Lp(a) levels should be described.
16) Bempedoic acid side effects- while the authors mention an increase in serum uric acid levels they do not discuss the increased risk of gout, which is a clinical concern.
17) Bempedoic acid and drug interactions- It is my understanding that bempedoic acid should not be used with simvastatin greater than 20 mg/day or pravastatin greater than 40mg/day.
18) One mentions costs but I think one should illustrate the differences in costs so readers can see the marked differences in price between statin and ezetimibe (both generic) and PCSK9 inhibitors and bempedoic acid.
Author Response
Comments to Authors
General Comments: I think the major problem is that it isn’t clear who this review is targeting. If it is targeting clinicians there are numerous deficiencies that are noted in my comments to authors. If is targeting pharmacologists there are also few areas that are not addressed. The authors tried to cover an enormous amount of information and therefore doesn’t provide sufficient information particularly as it relates to health care providers. The information presented seems most appropriate for pharmacologists. Perhaps if they were more focused the review would have been more valuable.
Specific Comments
1) “Currently, there are six statins available for clinical use: simvastatin, fluvastatin, atorvastatin, pravastatin, rosuvastatin, and pitavastatin”. There are actually seven statins available. Lovastatin is still available in the US.
Lovastatin has been added.
2) “These drugs slightly reduce triglycerides levels (10-30%)”. It should be noted that the effect of statins on TG levels is dependent on the patients TG level. The decrease in TGs occurs if TG levels are elevated. Additionally, the reduction in TGs is related to the decrease in LDL-C levels. The greater the reduction in LDL-C the greater the decrease in TGs (PMID: 9526817). This information is important for clinicians.
We thank the reviewer. The text now reads as follows “These drugs slightly reduce triglyceride levels (10-30%) but only in subjects with hy-pertriglyceridemia (i.e., the greater the LDL-C reducing efficacy is, the more effective the triglycerides reduction is) [15]. In addition to these actions, statins reduce apoB, non-HDL cholesterol, very-low density lipoprotein cholesterol (VLDL-C), VLDL triglycerides, and increase apoA-I and high-density lipoprotein cholesterol (HDL) by 5-10%”
3) One should mention the effect of statins on Lp(a) levels. As you are aware there is an increasing interest in Lp(a) as a causative factor in atherosclerosis.
We thank the reviewer. The text now reads as follows “Finally, it should be mentioned that an individual-patient-data analysis demonstrated that lipoprotein(a) levels increase significantly in patients started on statin therapy [2]. Specifically, the mean percent change from baseline ranged from 11.6% to 20.4% in the case of pravastatin and from 18.7% to 24.2% in the case of atorvastatin [3].”
4) In discussing the safety of statins ones needs a more extensive discussion. It is crucial that health care providers understand the details regarding the potential side effects of statins.
5) Muscle- A recent publication by the Cholesterol Treatment Trialists has presented detailed information on muscle complications based on a meta-analysis of 23 trials with over 100,000 participants (PMID: 36049498). Additionally, a number of studies have reported double blinded statin challenge studies in patients with “statin induced” muscle symptoms who were rechallenged with a statin vs. placebo (PMID: 25545331, 27039291). These and other studies should be discussed to put into perspective the effect of statins on muscle. It is essential that clinicians understand the relationship.
Diabetes- The frequency of this side effect should be mentioned and the risk factors for developing this complication discussed (PMID 20167359, 23219296).
Liver- The authors should point out that the difference in the risk of developing elevated transaminases in statin treated and controls is very small (1.4% in statin treated vs. 1.1% in controls; PMID 17159064) and that the FDA no longer recommends following liver function tests in patients treated with statins.
5) If this review is directed at pharmacologists one needs to discuss why gemfibrozil effects statin metabolism (Gemfibrozil inhibits the glucuronidation of statins, PMID: 24793440).
We thank the reviewer. The text now reads as follows “In particular, statin-associated muscle symptoms (SAMS) represent the most common symptoms that result in statin intolerance or statin discontinuation, increasing the risk for recurrent myocardial infarction (50%) and coronary heart disease events (51%) [24]. Specifically, statin intolerance is defined as one or more adverse effects associated with statin therapy, which resolves or improves with dose reduction or discontinuation. It can be classified as complete inability to tolerate any dose of a statin, or partial intolerance, with inability to tolerate the dose necessary to achieve the patient-specific therapeutic objective. To classify a patient as having statin intolerance, a minimum of two statins should have been attempted, including at least one at the lowest approved daily dosage [25]. However, data extrapolated from randomized clinical trials and real-world studies are contrasting on this issue (i.e., in long-term clinical trials, muscle adverse events occur at similar rates, regardless of allocation to statin or placebo). Indeed, trials with statins excluded many patients who would be prescribed a statin in clinical practice. These trials excluded patients unlikely to adhere to the demands of a long-term study and those taking known interacting drugs [26]. On this matter, an analysis of 19 placebo-controlled trials concluded that >90% of all reports of muscle symptoms by participants allocated statin therapy were not due to the statin [27]. Conversely, in routine care settings, this percentage can increase until 30% [28,29]. However, rechallenge is possible, by changing statin type and reducing the dose. As established by both interventional studies and clinical practice, it is recommended that before rechallenging, a washout period may be useful to allow symptom resolution, which varies among individuals [30]. A further approach can be the alternate-day dosing by prescribing atorvastatin and rosuvastatin which possess a long half-life [31].
Concerning new-onset diabetes, although statin therapy is associated with a 9% increased risk, it is low when compared with the reduction in coronary events. Indeed, treatment of 255 patients with statins for 4 years results in one extra case of diabetes [32,33]. Regarding the liver toxicity, the incidence of transaminase elevations >3-fold the upper limits of normal (ULN) have never been reported. This effect seems dose-related with a temporary pattern and, thus, clinicians should not be discouraged to prescribe statins even when a mild rise of serum liver enzyme is present [34].”
Development of cataracts or cognitive dysfunction and several more adverse events reported in the literature (e.g., proteinuria and haematuria) have been never confirmed to have a causal link with statin use [11]. The prevalence of complete statin intolerance might often be overestimated highlighting the need for the careful assessment of patients with potential symptoms related to statin intolerance. The excess rate of SAMS mainly derives from the so-called nocebo effect, as demonstrated in the ASCOT-LLA (An-glo-Scandinavian Cardiac Outcomes Trial--Lipid Lowering Arm) study. During the non-blinded non-randomised phase of this trial, SAMS were reported at a significantly higher rate by participants taking statins than by those who were not (Hazard ratio (HR)= 1.41 (95%CI 1.10-1.79) [35]. Indeed, muscle symptoms are referred more often when pa-tients and their doctors are aware that statin therapy has been used and not when its use was blinded.”
6) If this review is directed at clinicians one should focus on drugs that require an adjustment in the statin dose or make the use of statins contraindicated in the drug interaction section.
7) In the drug interaction section why no mention of amiodarone, diltiazem, and verapamil? These are commonly used drugs. Also, why no mention of grapefruit juice? Our patients often ask about this.
Gemfibrozil is also known to reduce the glucuronidation and elimination of statins.
Similarly, the dose of simvastatin should not exceed 20 mg per day when administered with the antiarrhythmic, P-gp inhibitor, amiodarone.
8) Interestingly, in the clinical efficacy section there is no mention of the populations where statins have failed to show clinical benefit. Specifically, in patients with renal disease on maintenance dialysis or congestive heart failure several studies have not shown benefits from statin treatment. It is essential that clinicians know when a drug will not be beneficial.
Despite guideline recommendations, among patients with established ASCVD, in the US fewer than one-half are statin users and only 22.5% are on a high-intensity formulation. In addition, in the case of end-stage renal disease, statins have shown no benefit on cardiovascular outcomes. Briefly, results of the AURORA (A study to evaluate the Use of Rosuvastatin in subjects On Regular hemodialysis: an Assessment of survival and cardiovascular events) study showed that rosuvastatin had no effect on nonfatal myocardial infarction, nonfatal stroke, or death from cardiovascular causes (HR= 0.96; 95%CI 0.84 to 1.11). The 4D (Die Deutsche Diabetes Dialyse) study conducted patients undergoing haemodialysis found the non-superiority of atorvastatin to reduce the composite primary end point of cardiovascular death, nonfatal myocardial infarction, and stroke.
9) In the ezetimibe section it is stated “The drug has shown, in combination with simvastatin, to reduce the risk of cardiovascular events in relation to the reduction of LDL-C both in nephropathic and in acute coronary syndrome patients”. It should be noted that there is a monotherapy trial where ezetimibe alone decreased cardiovascular events (PMID: 31434507).
However, in patients aged  75 years, ezetimibe, as a monotherapy, was able to reduce compared to usual care a composite of sudden cardiac death, fatal/nonfatal myocardial infarction, coronary revascularization, or fatal/nonfatal stroke by 34%.
10) The authors should emphasize that the combination of low dose statin + ezetimibe lowers LDL-C levels to almost the same extent as high dose statins (see table 2 of your paper). It is important for clinicians to know that one can, in certain patients, use a low dose of a statin with ezetimibe and achieve very good LDL-C reductions similar to intensive statin therapy.
Overall, this therapeutic approach leads to a LDL-C reduction similar to the one obtained with high intensity statins (i.e., 40 mg rosuvastatin or 80 mg atorvastatin).
11) The authors note that PCSK9 inhibitors decrease Lp(a) levels. The authors need to mention potential mechanisms for this observation as statins and ezetimibe, which also increase hepatic LDL receptors do not have this effect.
PCSK9 inhibitors lower plasma lipoprotein(a) concentrations by roughly 27%, with no significant difference between evolocumab and alirocumab. PCSK9 inhibitors increase the catabolism of lipoprotein(a) when used as a combination therapy or on a background of statin treatment in high-risk patients. However, as monotherapy, these agents mainly act through reduction of lipoprotein(a) production rate.
12) In the PCSK9 side effect section the following is stated “The most common adverse reactions reported in phase 3 clinical trials were muscle pain (27.2%), nasopharyngitis (9.3%), back pain (12%), upper respiratory tract infections (9%), headache (9.2%), joint pain (7%), flu-like symptoms (7.5%), or an increase in alanine or aspartate aminotransferase levels three times above the upper limit of normal (6%)”. Are these any different than what is observed in the control group? In studies one often sees common abnormalities, such as headaches, in a similar percentage of participants in the drug group and the placebo group.
The most common adverse reactions reported in phase 3 clinical trials were muscle pain, nasopharyngitis, back pain, upper respiratory tract infections, headache, joint pain (7%), flu-like symptoms, or an increase in alanine or aspartate aminotransferase levels three times above the upper limit of normal. However, no significant differences regarding the risk of adverse events leading to treatment discontinuation between alirocumab and evolocumab (relative risk= 0.98, 95% CI 0.79–1.22; P = 0.89) were found.
13) In the PCSK9 side effect section the following is stated “Most importantly, the PCSK9 inhibition by evolocumab and alirocumab is not associated to higher risk of new cases of diabetes”. It should be noted that both the FOURIER and ODISSEY OUTCOMES were both relatively short trials and therefore could have missed an increase in diabetes. Note it took many years and many trials before the link between statins and diabetes were recognized and individual trials often did not show an increase. If you examine figure 2 in the classic meta-analysis of statin induced diabetes (PMID: 20167359) you will see that the only study that showed a statistically significant increased risk of diabetes was the JUPITER Trial and several actually showed a reduced risk (non-significant). It should be noted that a Mendelian randomization study of PCSK9 variants has raised the possibility of a link with diabetes (PMID: 27908689). Thus, I think further studies are needed. I think it is essential to carefully review the literature before making definitive statements that are clinically very important.
Relative to the risk of developing new-onset diabetes, evolocumab and alirocumab were primarily related to mild hyperglycemia rather than diabetes, with most of the hyper-glycemic events occurring during the first 6 months of treatment. Thus, a careful monitoring and identification of individual more susceptible to develop diabetes is advised.
15) The effect of bempedoic acid on Lp(a) levels should be described.
Overall, bempedoic acid had no clinically important impact on lipoprotein(a).
16) Bempedoic acid side effects- while the authors mention an increase in serum uric acid levels they do not discuss the increased risk of gout, which is a clinical concern.
Relative to new-onset diabetes, data of the CLEAR Outcomes showed that the proportion of patients who developed new-onset diabetes were similar between the bempedoic acid and placebo groups, with 429 of 3848 (11.1%) with bempedoic acid versus 433 of 3749 (11.5%) with placebo (HR= 0.95; 95%CI 0.83-1.09) [132]. The same trial highlighted a higher incidence of cholelithiasis (2.2% vs. 1.2%) in the bempedoic acid group compared to placebo. This should be related to the inhibition of the OATP1B1/3 expressed mainly in hepatocytes.
18) One mentions costs but I think one should illustrate the differences in costs so readers can see the marked differences in price between statin and ezetimibe (both generic) and PCSK9 inhibitors and bempedoic acid
We believe this is out of the scope of the review.
Reviewer 3 Report
Comments and Suggestions for Authors
The narrative review by Ferri et al summarize the currently available lipid-lowering therapies, starting with the classical statins up until the novel bempedoic acid. The manuscript is decently written (albeit without novelties), summing up the particularities of these molecules. Some aspects should be further clarified:
- I don’t consider essential to depict the chemical structures of all the statins, in fact it would be appropriate to show in a figure the mechanisms of the novel molecules (inclisiran, bempedoic acid).
- Moreover, it would be better to detail (or display in a figure) the mechanisms related to the pleiotropic effects of statins (essentially important in acute coronary syndromes)
- Please detail the management of statins adverse effects, such as myositis/rhabdomyolysis, and elevated liver enzymes, according to different cut-offs.
- I recommend more references related to the prognostic role of those molecules (not only related to the decreased LDL-C).
- What is the impact of inclisiran and bempedoic acid on triglycerides? What about Lp(a) or ApoB?
Best regards,
The Reviewer
Comments on the Quality of English LanguageEnglish seems fine
Author Response
The narrative review by Ferri et al summarize the currently available lipid-lowering therapies, starting with the classical statins up until the novel bempedoic acid. The manuscript is decently written (albeit without novelties), summing up the particularities of these molecules. Some aspects should be further clarified:
- I don’t consider essential to depict the chemical structures of all the statins, in fact it would be appropriate to show in a figure the mechanisms of the novel molecules (inclisiran, bempedoic acid).
Response. we agree with the reviewer. we have erased the chemical structures of statins and ezetimibe. However, we believe that bempedoic acid reserves more attention since it is the newest small molecule drug. We, therefore, decided to leave its chemical structure. As indicated by a second reviewer, we have added a new figure showing the mechanism of action of these drugs.
- Moreover, it would be better to detail (or display in a figure) the mechanisms related to the pleiotropic effects of statins (essentially important in acute coronary syndromes).
Response: although we agree with the reviewer that this is a very important topic. We believe that it would be better to not emphasized this pharmacological property of statins. Indeed, there are still a lot of debate on their clinical evidence.
- Please detail the management of statins adverse effects, such as myositis/rhabdomyolysis, and elevated liver enzymes, according to different cut-offs.
Response: we thank the reviewer for the suggestion. This is a very important issue. However, we believe that it goes beyond the scope of our review. The description of the current guidelines for the management of statin-side effects required an extensive work that potentially can be described in a dedicated consensus paper. We, therefore, decided to not describe this issue in the present manuscript. We hope that the reviewer will be in agreement with our decision.
- I recommend more references related to the prognostic role of those molecules (not only related to the decreased LDL-C).
Response: These have been added.
- What is the impact of inclisiran and bempedoic acid on triglycerides? What about Lp(a) or ApoB?
Response: We have now added the following sentence: “Thus, bempedoic acid reduces LDL-C (20-25%), non-HDL-C (19%), apo B (15%) and total cholesterol (16%) in patients with hypercholesterolemia or mixed dyslipidemia [1], while no effect is observed on TG levels (+0.4%)[120], hs-CRP [121].
120) Laufs, U.; Banach, M.; Mancini, G.B.J.; Gaudet, D.; Bloedon, L.T.; Sterling, L.R.; Kelly, S.; Stroes, E.S.G. Efficacy and Safety of Bempedoic Acid in Patients With Hypercholesterolemia and Statin Intolerance. Journal of the American Heart Association 2019, 8, e011662, doi:10.1161/JAHA.118.011662.
121) Ridker, P.M.; Lei, L.; Ray, K.K.; Ballantyne, C.M.; Bradwin, G.; Rifai, N. Effects of bempedoic acid on CRP, IL-6, fibrinogen and lipoprotein(a) in patients with residual inflammatory risk: A secondary analysis of the CLEAR harmony trial. Journal of clinical lipidology 2023, 17, 297-302, doi:10.1016/j.jacl.2023.02.002.
Round 2
Reviewer 2 Report
Comments and Suggestions for Authors
None
Comments on the Quality of English LanguageOkay